# Daily vocal exercise is necessary for peak performance singing in a songbird

Iris Adam [1] ✉, Katharina Riebel [2], Per Stål[3], Neil Wood[4], Michael J. Previs[4] & Coen P. H. Elemans [1] ✉

Vocal signals, including human speech and birdsong, are produced by complicated, precisely coordinated body movements, whose execution is fitness-determining in resource competition and mate choice. While the acquisition and maintenance of motor skills generally requires practice to develop and maintain both motor circuitry and muscle performance, it is unknown whether vocal muscles, like limb muscles, exhibit exercise-induced plasticity. Here, we show that juvenile and adult zebra finches (*Taeniopygia castanotis*) require daily vocal exercise to first gain and subsequently maintain peak vocal muscle performance. Experimentally preventing male birds from singing alters both vocal muscle physiology and vocal performance within days. Furthermore, we find females prefer song of vocally exercised males in choice experiments. Vocal output thus contains information on recent exercise status, and acts as an honest indicator of past exercise investment in songbirds, and possibly in all vocalising vertebrates.

Producing complex learned vocalizations, such as human speech and birdsong, comprises some of the most intricate, temporally precisely coordinated movements of the vertebrate body. The precise motor control and execution of these vocal motor skills ultimately plays a decisive role in mate choice and resource competition[1,2]. The acquisition and maintenance of motor skills requires motor practice i) to develop and maintain motor circuitry and ii) to improve and maintain muscle performance. In vocal learners, vocal skills are typically acquired over postnatal development, when the body is growing, and maintained over adulthood. Especially for songbirds – widely accepted as the closest animal analogue for human speech acquisition[3,4] – the critical contribution of reshaping sensory and motor circuits to song learning is well established[3,5–8]. However, whether vocal muscles require motor practice to improve and maintain performance, and if such changes directly affect vocal output remains unknown for any vocalizing vertebrate, including humans[9].

Skeletal muscles show life-long plasticity in response to changes in functional load or hormonal state, which has been studied extensively in limb muscles[10,11]. Unloading paradigms, such as disuse, denervation and bed rest, cause mammalian limb muscles typically to transform slow fibre types into fast fibre types, while reversely loading paradigms, such as training programs and electrical stimulation, typically transform fast into slower fibre types[12]. However, myoblast lineage, postnatal motor activation patterns and exercise play crucial roles in the development of muscle groups or allotypes[13]. The craniofacial muscles that include extraocular, jaw, most laryngeal and syringeal muscles are considered distinct allotypes from limb muscles, particularly for their ability to express specialized myosin isoforms[10]. Additionally, craniofacial and limb muscles differ in their response to stimulation and disuse paradigms as well as neuromuscular diseases[13,14]. In humans, the larynx and its muscles are hypothesized to change with training or age, but in vivo experiments are challenging or impossible[9]. In songbirds, the muscles controlling the avian vocal organ, the syrinx, have been established to change concurrent with song learning during postnatal development[14–16]: both mass and speed of vocal muscle gradually increase over this period[14–16] until they attain the fastest contraction kinetics of any vertebrate muscle in adulthood[14]. These postnatal muscle changes have been hypothesised to arise from either muscle maturation or brain-body interaction via exercise-induced plasticity[17], but we currently do not know if vocal

[1]Department of Biology, University of Southern Denmark, Odense, Denmark. [2]Institute of Biology, Animal Sciences & Health, Leiden University, Leiden, The Netherlands. [3]Department of Integrative Medical Biology, Umea University, Umeå, Sweden. [4]Department of Molecular Physiology and Biophysics, Larner College of Medicine, University of Vermont, Burlington, NJ, USA. ✉e-mail: irisadam@biology.sdu.dk; coen@biology.sdu.dk

muscles exhibit exercise-induced plasticity for key performance parameters such as speed and force. Moreover, the question whether vocal muscles need exercise to acquire peak performance has remained overlooked and untested.

A regular need to exercise vocal muscles for maintaining their peak performance is consistent with the daily singing behaviour of many bird species. The main function of this daily song is considered to tie and maintain social bonds as well as defend territories[2]. However, many birds also sing daily outside of these contexts: in captivity, even isolated zebra finches sing hundreds of songs per day[18,19], and in the wild birds keep singing daily even under extremely adverse conditions[20]. If birds need to sing regularly for exercise to keep their muscles in shape, this could contribute to explain why they sing even in situations where it would not aid territorial defence, mate attraction or other functions. Moreover, if syringeal muscles exhibit exercise-induced plasticity, vocal performance may include signatures of recent exercise and sound output may relay vocal exercise history to listeners.

Here we show that initial acquisition and subsequent maintenance of adult peak vocal muscle performance requires vocal muscle exercise in a well-studied vocal learner - the zebra finch. We furthermore show that exercise-induced muscle property changes affect vocal production and that conspecifics can detect and evaluate these changes in a mate-choice context.

## Results

### Exercise is needed to gain and keep vocal muscle performance

First, to test the hypothesis that exercise is needed to acquire peak adult vocal muscle performance, we exploited the bipartite structure of the songbird syrinx where two bilateral pairs of vibrating labia (left and right hemisyrinx) independently contribute to song[21]. We allowed normal muscle use during vocal ontogeny of the left hemisyrinx (i.e., intact side as control), and experimentally prevented use of the right hemisyrinx by muscle denervation in juveniles (Fig. 1; See Methods). Next, we measured two key features of physiological muscle performance in the dorsal tracheosyringeal muscle: i), contraction speed, measured as the full width at half maximum (FWHM) of force development during twitch contraction, and ii), maximal contraction force per cross-sectional area, i.e. the maximal isometric stress (MIS) during tetanic contraction (Fig. 1b). We compared these muscle features on both sides when these animals were adult (100 days post hatching) and song learning was completed. Denervation halved the contraction speed (FWHM doubled from $4.89 \pm 0.86$ ms to $9.13 \pm 3.18$ ms, unpaired two-sided Welch's t-test, $p = 0.00208$, $N = 7$), quadrupled MIS from $2.61 \pm 1.37$ mN/mm$^2$ to $12.23 \pm 9.36$ mN/mm$^2$ (Fig. 1b; unpaired two-sided Welch's t-test, $p = 0.01508$, $N = 7$), and reduced cross-sectional area (CSA) by 70% (Supplementary Fig. 1a, b, paired two-sided Welch's t-test, $p = 0.00241$, $N = 4$). The myofibre number remained the same (Supplementary Fig 1c, paired two-sided Welch's t-test, $p = 0.04385$, $N = 4$). Treatment did not affect the intact side, because both contraction speed and MIS were not different from untreated adult males (Fig. 1b, Supplementary Data 1). The large treatment-induced differences in physiological performance were also evident in muscle morphology: the denervated adult hemisyrinx muscle visually resembled the juvenile syrinx at 25 days old (Fig. 1c). Taken together, our data support the hypothesis that achieving adult peak muscle performance requires vocal muscle exercise during the song-learning phase.

Second, to test whether continued vocal muscle exercise is needed to maintain adult performance, we prevented muscle exercise in adult males by unilateral denervation (Fig. 1d, See Methods). Muscle speed decreased rapidly on the denervated side, while it remained unaffected on the intact side. Only two days postdenervation the syringeal muscle had slowed down, and after 21 days postdenervation its speed reverted to juvenile levels (Fig. 1e). The effects of denervation on MIS were even more dramatic: merely two days postdenervation MIS dropped fivefold compared to the intact side (from $7.13 \pm 4.85$ to $1.43 \pm 1.88$ mN/mm$^2$, unpaired Welch's t-test, $p = 0.00148$, Fig. 1f), which remained unaffected compared to untreated males. Morphologically, denervation reduced the CSA to $74 \pm 8\%$ (Paired two-sided Welch's t-test, $p = 0.01128$, $N = 3$) of the intact side 21 days postdenervation (Fig. 1g) while the number of muscle fibres remained the same (Fig. 1h, Paired two-sided Welch's t-test, $p = 0.9431$, $N = 3$). Thus, denervation severely affects key physiological and morphological muscle features that cause these vocal muscles to lose their peak performance within days. Taken together, these data support the hypothesis that maintaining adult peak muscle performance requires vocal muscle exercise.

### Disuse changes vocal muscle anatomy and protein expression

Laryngeal and limb muscles contain multiple distinct fibre types with different features pertaining to force production, fatigue resistance and energy metabolism[10]. In zebra finch males the majority (67–87%) of all muscle fibres is classified as superfast muscle fibres[14,22] that are not immunoreactive to any available antibody raised against heavy myosin chain (MyHC) isoforms. The remaining 13–33% of syringeal muscle fibres are smaller diameter fibres immunoreactive to an antibody binding to mammalian fast twitch MyHCs (MY-32)[14,22]. To quantify if and how the two syringeal muscle fibre types – superfast and fast - are affected by disuse, we prevented use by unilateral denervation in adult males, and measured fibre CSA and MY-32 expression intensity (Fig. 2a–c; See Methods). On the intact side, we observed $66 \pm 3\%$ ($N = 3$) unstained fibres of varying size (200–1000 μm$^2$) intermingled with 34%, smaller (100–400 μm$^2$) fibres of varying MY-32 intensity (Fig. 2a, c), as in wild type males[14,22]. The binomial fibre area distribution combined with a unimodal MY-32 intensity distribution (Fig. 2c) shows two fibre types, corroborating earlier results[14,22]. Denervation increased fibre MY-32 intensity, decreased fibre CSA and resulted in a unimodal fibre CSA distribution (Fig. 2c, d). The total number of superfast fibres decreased (Fig. 2d, paired two-sided t-test, $p = 0.0088$, $N = 3$) and superfast fibre fraction reduced from $66 \pm 3\%$ to $44 \pm 4\%$. Syringeal muscle disuse thus drives fibre type composition towards fibres with smaller CSA and slower contractile properties.

The physiological and fibre typing data combined suggest that syringeal muscle fibres respond to disuse with both quantitative and qualitative changes in protein expression: reduced fibre CSA and MIS suggest reduced protein abundance, and reduced speed combined with increased reactivity to MY-32 suggest qualitative expression changes of MyHC isoforms. To identify and quantify disuse-induced protein expression changes, we performed proteomic profiling using liquid chromatography mass spectrometry first in untreated syringeal muscles (See Methods). We categorized the identified proteins according to their role in 1) force production (sarcomeric), 2) calcium handling and 3) mitochondrial function. We identified 450 proteins with high confidence (Supplementary Data 2) of which over 80% were sarcomeric, calcium handling and mitochondrial proteins (Fig. 2e, f, Supplementary text). We detected five MyHC isoforms with MYH13 as the most abundant ($92 \pm 3\%$) of the total myosin pool (Fig. 2h). Expression of myosin light chains (Fig. 2i), troponin subunits and calcium handling proteins were identical to mammalian fast twitch isoforms. Interestingly, parvalbumins – cytosolic calcium buffers - were expressed an order of magnitude higher than in mouse limb muscle[23] (Fig. 2j), which aids fast muscle relaxation during bouts of muscle activity by temporarily sequestering calcium and pumping it into the sarcoplasmic reticulum asynchronously between bouts of activity[24]. Mitochondrial proteins were highly abundant (Fig. 2k), making up $38 \pm 6\%$ of the total proteome.

Next, we quantified protein expression changes after 21 days of disuse by denervation. Total protein abundance of all three functional categories reduced, except for uncategorized proteins (Fig. 2f). Expression changed significantly in 175 proteins, with the majority (150/175) decreasing in abundance (Fig. 2g, Supplementary Data 1).

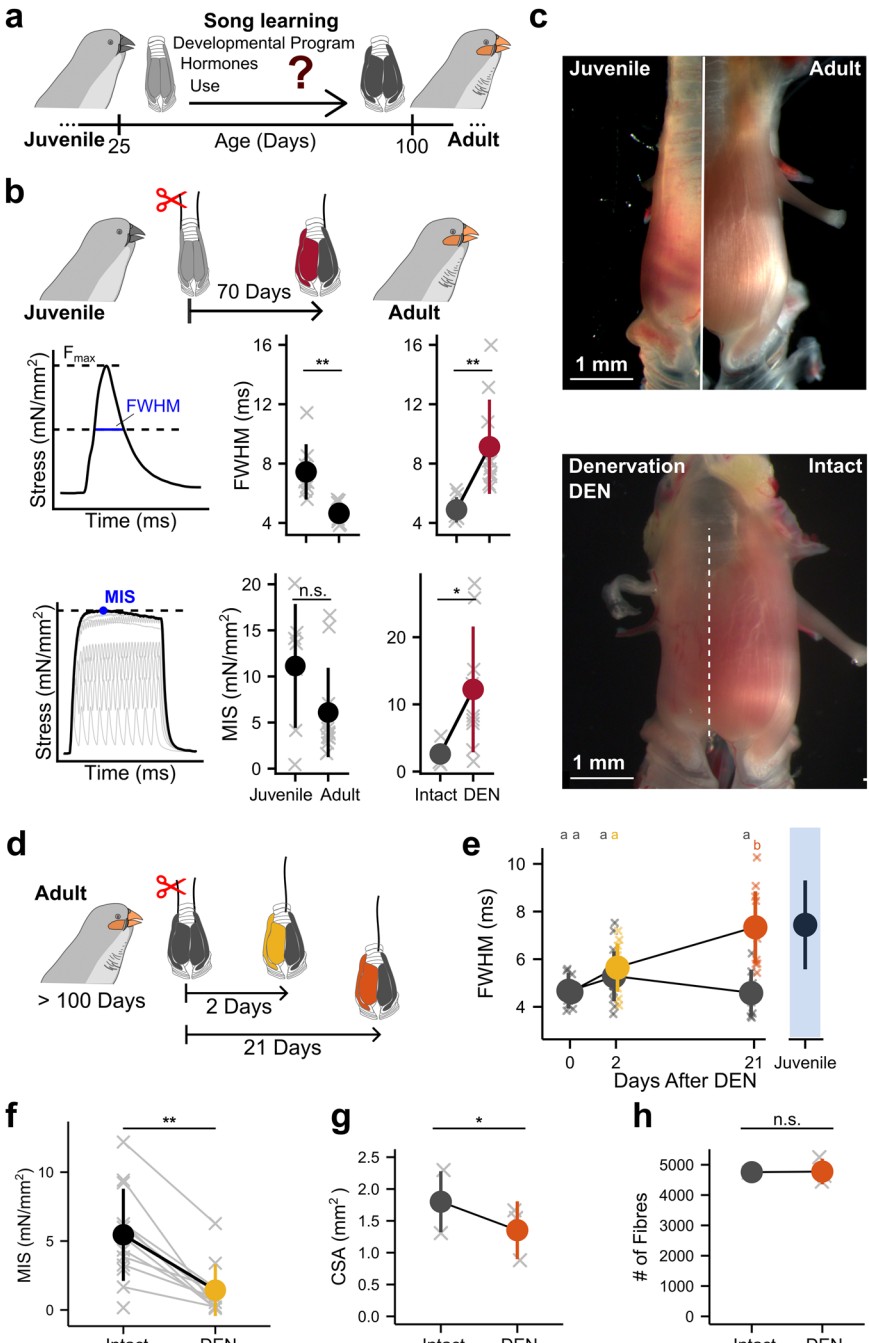

**Fig. 1 | Acquisition and maintenance of adult vocal muscle performance requires exercise. a** During song learning, vocal muscles in male zebra finches gradually hypertrophy, **b**, increase in contraction speed (Unpaired two-sided Welch's t-test, $p = 0.00345$, $N_{Adults} = 11$, $N_{Juveniles} = 8$) and decrease in maximal isometric stress MIS (Unpaired two-sided Welch's t-test, $p = 0.114$, $N_{Adults} = 12$, $N_{Juveniles} = 7$) from juvenile to adult. Denervation (DEN) of the right hemisyrinx at the onset of song learning (dark red) decreases adult contraction speed (Unpaired two-sided Welch's t-test, $p = 0.00208$, $N_{Intact} = 7$, $N_{DEN} = 9$), measured as the full width at half maximum (FWHM) of isometric force development of twitch contractions. Maximal isometric stress (MIS) during tetanic stimulation is increased compared to the intact side (grey) (Unpaired two-sided Welch's t-test, $p = 0.01508$, $N_{Intact} = 7$, $N_{DEN} = 9$). **c** Muscles on the denervated hemisyrinx remain small and resemble juvenile syrinx muscles. **d** In adult males, vocal muscle

performance was measured 2 (yellow) and 21 days (orange) after denervation (DEN) on the denervated and intact control (grey) side. **e** FWHM increased and after 21 days reverted to juvenile speed (black) (Mixed-effects model: FWHM~treatment*Days_after_DEN + (1|bird), with pairwise comparison with Tukey's adjustment, $N_{Day0} = 11$, $N_{Day2,Intact} = 16$, $N_{Day2,DEN} = 11$, $N_{Day21,Intact} = 8$, $N_{Day21,DEN} = 12$). **f** MIS reduced fivefold two days after denervation (Unpaired two-sided Welch's t-test, $p = 0.00148$, $N_{Day2,Intact} = 16$, $N_{Day2,DEN} = 11$). **g** Muscle cross-sectional area (CSA) was significantly reduced 21 days after denervation (Paired two-sided Welch's t-test, $p = 0.01128$, $N = 3$), but **h**, muscle fibre number did not change (Paired two-sided Welch's t-test, $p = 0.9431$, $N = 3$). * at $p < 0.05$, ** at $p < 0.01$, n.s. at $p \geq 0.05$. Unshared lower-case letters in e indicate post hoc differences with a significance level $p < 0.05$. Data are presented as mean values $\pm 1$ S.D. Source data are provided as a Source Data file.

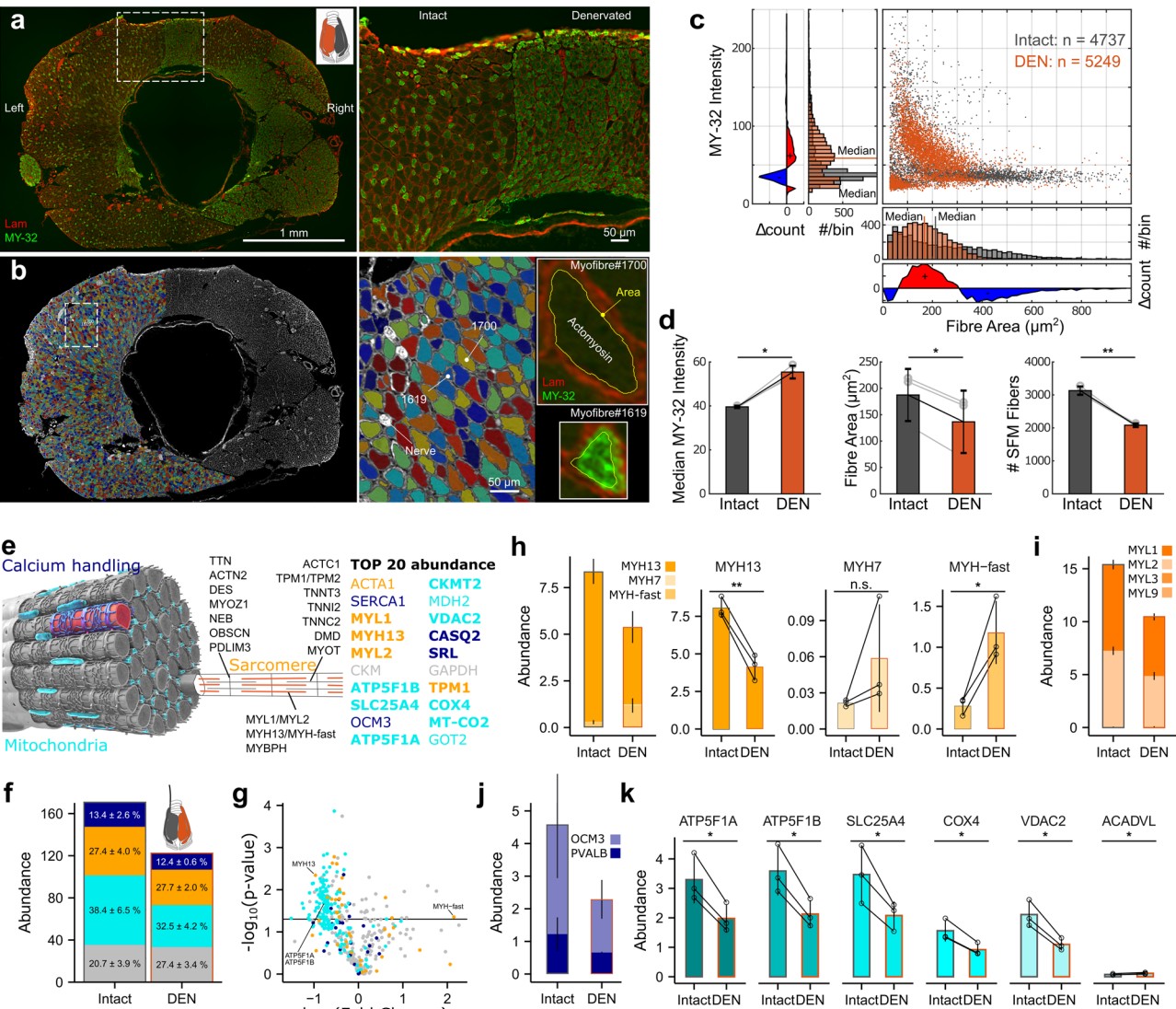

**Fig. 2 | Disuse changes fibre type organization and protein expression in vocal muscles. a** Cross-section (left) and detail (right) of syringeal muscle fibres of adult male syrinx immunostained for laminin (Lam) and fast MyHCs (MY-32) after 21 days unilateral denervation (DEN). **b** Automated myofibre detection shown in the intact hemisyrinx, faux-color-coded for fibre ID. Right, detail with (top) a typical superfast myofibre without MY-32-reactivity (myofibre #1700), and (bottom) a typical fast fibre with high MY-32 reactivity (myofibre #1619). Images are representative of results obtained from 3 independent animals. **c** After denervation (orange), fibre type composition shifts from large superfast fibres (MY-32 negative) with interspersed small fast (MY-32 positive) fibres to medium-sized fast fibres. **d** Denervation increases median MY-32 reactivity (Paired two-sided t-test, $p = 0.012$, $N = 3$, independent animals), reduces median fibre area (Paired two-sided t-test, $p = 0.012$ $N = 3$ independent animals) and reduces the number of superfast fibres

(Paired two-sided t-test, $p = 0.0088$, $N = 3$ independent animals). **e** Schematic of superfast syringeal muscle fibre with identified proteins and top 20 most abundant proteins color-coded for categories; sarcomeric (orange), mitochondrial (cyan) and calcium handling (blue). Bold type are proteins significantly affected by denervation (See Supplementary Data 1). **f** Disuse by denervation causes downregulation of total protein abundance. **g**, Volcano plot of 450 identified proteins showing statistical significance over magnitude of fold change due to denervation.
**h**, Abundance of total MyHC proteins, **i**, myosin light chains, **j**, parvalbumins, and **k**, mitochondrial proteins decrease after denervation (Changes in gene expression were tested using Paired two-sided Welch's t-test, see Supplementary Data 1, $N = 3$). Data are presented as mean values ± 1 S.D. * at $p < 0.05$, ** at $p < 0.01$, n.s. at $p \geq 0.05$. Source data are provided as a Source Data file.

Denervation affected the expression of 54.4% of sarcomeric, calcium handling and mitochondrial proteins, but only 20% of uncategorized proteins (Fig. 2g). The overall abundance of MyHCs was decreased to 64.3 ± 1.8% of the intact side driven by decreased MYH13 expression (96.3 ± 1.5% to 76.5 ± 10.0%). The expression of fast MyHC isoforms increased at the same time (MYH-fast 3.5 ± 1.5% to 22.4 ± 9.1%), shifting MyHC composition (Fig. 2h). Because MYH13 remained highly abundant, MY-32 reactive fibres must co-express MYH13 after denervation (Fig. 2a, b). Expression of all calcium handling proteins, including parvalbumins (Fig. 2j) and all but one mitochondrial protein (ACADVL) decreased (Fig. 2k). Thus, denervation reduced overall abundance of protein categories that set muscle speed, such as calcium handling and

mitochondrial function, and drove composition change of MyHCs from fast to slow isoforms, consistent with our morphological and physiological data.

### Singing prevention changes muscle and vocal performance

Syringeal vocal muscles are active during song production, but also during calling[25] and even rhythmically to open airways during breathing[26], which may aid to upkeep muscle performance. To test specifically if singing rather than calling is required to maintain peak muscle performance in adults, we prevented adult males from singing while retaining all other functions (Fig. 3a, See Methods). After seven days without singing, vocal muscles were significantly slower than

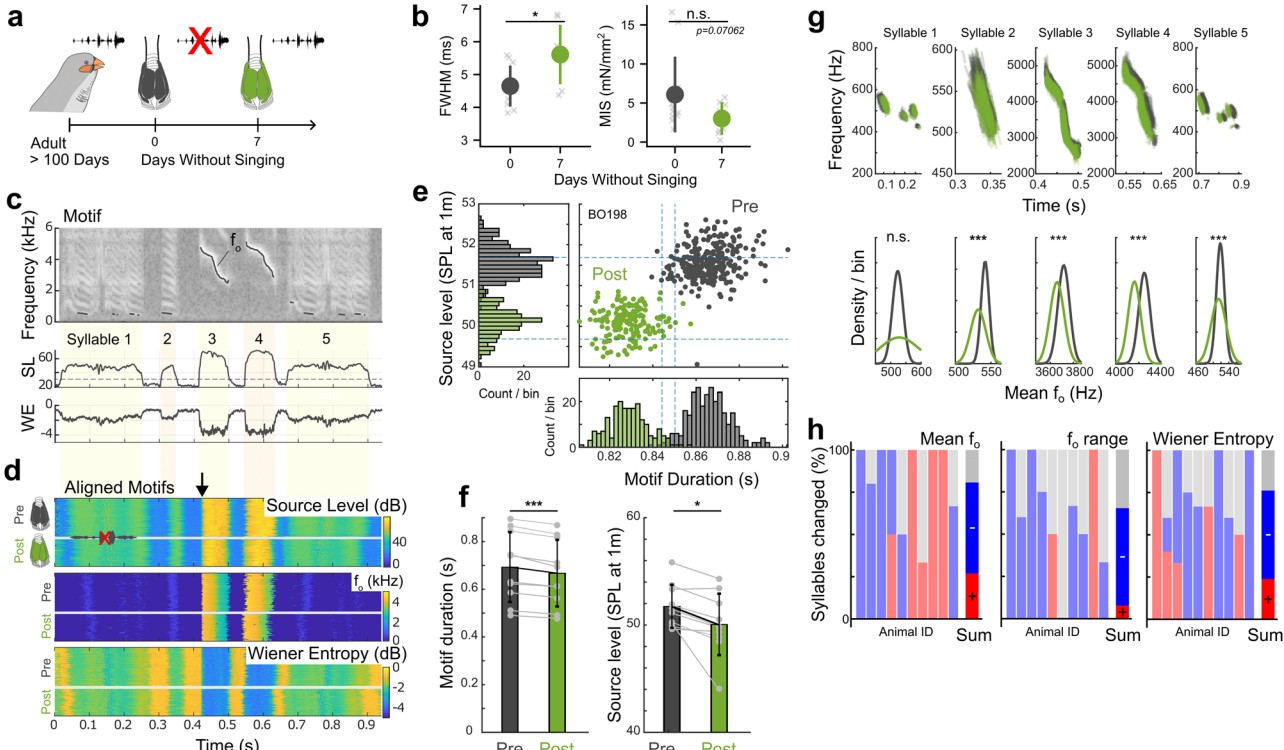

**Fig. 3 | Singing exercise drives changes in vocal output. a** After seven days of singing prevention (green) **b,** muscle speed was significantly slower compared to singing adult males (Unpaired two-sided Welch's t-test, $p = 0.02409$, $N_{Adults} = 11$, $N_{SingingPrevention} = 8$) and MIS was reduced, albeit not significantly (Unpaired two-sided Welch's t-test, $p = 0.07062$, $N_{Adults} = 12$, $N_{SingingPrevention} = 8$). **c** Example of stereotyped song motif (individual BO198). Top to bottom: spectrogram, source level (dB$_{rms}$ re. 20μPa at 1 m) and Wiener entropy (dB). Fundamental frequency ($f_o$) trace is overlaid on spectrogram. **d** Motif iterations pre ($n = 284$) and post ($n = 201$) singing prevention of same individual aligned to the onset of syllable three (downward arrow). Each row represents a single motif iteration color-coded for source level (top), $f_o$ (middle) and Wiener entropy (bottom). **e** Individual BO198 and **f**, group data show that motif source level (Paired two-sided t-test, $p = 0.011$, $N = 10$ independent animals) and duration (Paired two-sided t-test, $p = 7.7*10^{-5}$, $N = 10$ independent animals) decreased significantly due to singing prevention. **g** $f_o$ traces (top) and mean $f_o$ distributions (bottom) per syllable of individual BO198 and **h,** all individuals show that singing prevention causes significant changes in $f_o$, $f_o$-range and Wiener entropy in 81% (21/26), 65% (17/26), and 76% (32/42) of all analysed syllables of 10 animals respectively. (Red: significant increase ($p < 0.05$), blue: significant decrease ($p < 0.05$), grey: no statistically significant change ($p \geq 0.05$), unpaired two-sided t-test). Data are presented as mean values ± 1 S.D. * at $p < 0.05$, *** at $p < 0.001$, n.s. at $p \geq 0.05$. Source data are provided as a Source Data file.

control muscles of males allowed to sing freely (Fig. 3b, unpaired two-sided Welch's t-test, $p = 0.02409$). Furthermore, MIS halved, but this effect was not significant (Fig. 3b, unpaired two-sided Welch's t-test, $p = 0.07062$). Additionally, we quantified changes in protein abundance and composition and found that the protein abundance of sarcomeric, mitochondrial, as well as calcium handling proteins was reduced, while the composition of the MyHC-pool was less affected (Supplementary Fig. 2). Because the short-term singing prevention paradigm affects speed, MIS and proteome in the same direction as short-term denervation, it acts as a milder unloading paradigm compared to nerve cuts. These data show that breathing and calling activity alone is insufficient, and adult males need regular singing exercise to maintain peak vocal muscle performance.

To test whether singing prevention-induced muscle changes drive acoustic changes in song, we compared song before and one week after targeted singing prevention (See Methods). Singing prevention caused several acoustic changes to song. First, motif duration decreased from $0.69 \pm 0.15$ to $0.67 \pm 0.14$ s (Paired two-sided t-test, $p = 7.7*10^{-5}$, $N = 10$), i.e. decreasing by $3.7 \pm 1.4\%$ and motif source level decreased from $51.7 \pm 2.0$ to $50.1 \pm 2.9$ dB SPL at 1 m (Fig. 3c–f, paired two-sided t-test, $p = 0.011$, $N = 10$). Second, we extracted time-resolved fundamental frequency ($f_o$) traces over individual syllables within motifs (Fig. 3g). Although the overall shape of these $f_o$ trajectories did not change, the mean $f_o$ changed significantly in 81%, $f_o$ range in 65% and Wiener entropy in 76% of syllables from 10 animals (Fig. 3h). Taken

together, our acoustic analyses showed that only one week of our targeted singing prevention paradigm causes significant shifts in vocal output.

## Females prefer exercised song

Lastly, we tested whether the acoustic changes observed after singing prevention were perceived by and meaningful to females to whom males direct song for mate attraction. Females were exposed to playback of songs from one male in a validated operant song preference test[27]. By pecking operant keys, female zebra finches could trigger playbacks of either pre- or post-singing prevention songs from the same male (Fig. 4a, see Methods). Within a test, the two stimulus songs had the same motif duration and source level[28–30]. Females were offered only one iteration of the recorded songs which means that to discriminate between songs, females had to detect the within motif acoustic changes, rather than variability among motifs or singing speed (parameters shown to positively affect choice[31]). Even with these cues removed, eight out of nine females showed a significant preference for a song (Fig. 4a, b, G-test with William's correction, Supplementary Data 1). Of the eight females, the majority (75%) preferred song from before singing prevention over song after singing prevention (Fig 4c, d). Females can thus perceive and distinguish the treatment-induced changes in muscle performance by listening to a single iteration of a male's song. Importantly, they prefer songs produced by exercised males.

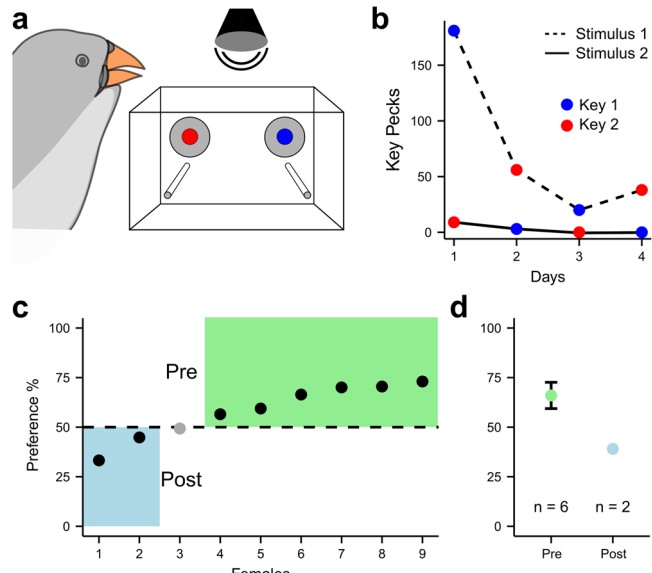

**Fig. 4 | Females choose and prefer song of exercised males. a** Female choice preference test setup to elicit playback of song pre and post short-term singing prevention. **b** Key peck events per day (female 6), where stimuli are side-switched daily to control for side-bias. **c** Of the 9 tested females, 8 significantly preferred (black) one stimulus above chance (G-test with William's correction, See Supplementary Data 1). Six (75%) preferred the exercised song (pre singing prevention), **d** with a mean preference of 66.0 ± 6.62%. Data are presented as mean values ± 1 S.D. Source data are provided as a Source Data file.

## Discussion

Combined, our physiological, morphological, molecular, and behavioural data show that juvenile and adult male zebra finches need to sing daily to gain and maintain peak vocal performance. Muscle exercise rapidly alters vocal output and conspecific females detect and prefer exercised songs. Conspecific receivers can thus use vocal performance as a proxy for the sender's recent (and possibly long-term) exercise investment. The requirement of daily exercise to maintain peak performance provides a mechanistic explanation for song as an honest signal for the sender's condition. In most vocalizing tetrapod species, muscles either precisely control the vocal organ, respiratory system, and vocal tract to modulate flow-induced vibrations or directly produce acoustic signals through contractions[32]. Neuromuscular effects of individual exercise history on vocal output are thus likely extendable to all vocalizing vertebrates.

We show that vocal exercise to optimize muscle performance is an integral part of vocal skill learning in songbirds. During song learning, the song system changes profoundly in size, number of neurons, connectivity and firing properties[5]. Up to the end of this period, syringeal muscles keep changing in weight, MIS and speed[14–16] due to exercise. Thus, both circuit remodelling[5] and vocal muscle exercise combine to achieve the precision of final song execution and the duration of both processes contributes to and constrains the duration of vocal skill learning. What neural stimulation patterns in song specifically promote muscle hypertrophy and speed increase needs further investigation, but a possible mechanism is bursts of high-frequency motor neuron firing - as observed in premotor neurons[33] - increase expression of faster MyHCs and mitochondrial content through IGF1 signalling. Interestingly, in limb muscles, first-time muscle training affects subsequent re-training duration by increasing myonuclei in existing muscle fibres[34]. These myonuclei are retained during subsequent muscle atrophy and supposedly serve as cellular muscle memory to allow faster muscle performance gains when retraining[34]. This process could explain how seasonally singing

songbirds reach peak performance faster in their second year[35]. Thus, early exercise can impact the speed of future skill learning with lifelong consequences.

To prevent such a complex behaviour as singing, prevention paradigms risk affecting birds' physiology and behaviour in other ways. Birds were temporarily housed in dark conditions which might not only have changed the amount of vocal exercise, but also affected motivation to vocalise, increased stress and altered melatonin levels, all of which are known to drive changes in song acoustics[36–41]. Behavioural observations suggested that, if anything, our birds had a raised motivation to sing, because we had to prevent them from singing during feeding sessions and they immediately started singing when the lights were turned on after the paradigm, consistent with earlier findings[39,40]. During our behavioural observations we did not detect signs of severe stress (reduced mobility, reduced food intake), the birds did not lose weight and were eager to sing throughout. Furthermore, songbirds that were prevented from singing by weights around their necks showed acoustic changes in line with the changes reported here independent of concurrently measured stress indicators corticosterone level or body weight[42]. Thus – as here - song changes most likely must have resulted from to the lack of singing activity[42]. Likewise, decrease of melatonin signalling - due to changes in lighting conditions[36], pharmacological[36,37] or surgical intervention[36] – decreases motif duration but not spectral features of song[36], making melatonin an unlikely explanation of our observations. Reversely, increased melatonin levels increase motif duration in zebra finches[36] and efficiently counteract muscle atrophy in humans and mammalian animal models[43,44], and is thus not consistent with the changes we observed in muscle physiology. Although we do not exclude other factors from playing a role, we think that our song prevention paradigm predominantly supressed singing activity.

Our singing prevention paradigm caused song parameter changes, of which only some we exclusively attribute to vocal muscle properties. (*i*) Fundamental frequency and WE. Changes in force produced by syringeal muscles through electrical stimulation affects the acoustic parameters fundamental frequency and Wiener entropy ex vivo and in vivo[45–47], thus reducing MIS will also affect them. In vivo, these acoustic parameters are set by several control parameters in the syringeal, respiratory, and vocal tract motor systems[46,48] and we currently have insufficient mechanistic insight into how individual syringeal muscles affect acoustic parameters[45,49]. Vocal muscle activity correlates with acoustic features such as $f_o$ within some syllables[25], but the correlation strength and polarity depend on the syllable[45,49]. Our data further supports this idea, because the strength and polarity of the acoustic effect caused by reduced MIS and twitch speed of the vocal muscles are significant but also individual and syllable dependent. (*ii*) Motif duration is predominantly set by temperature of the motor pathway[50]. The 3.7% motif duration decrease we observed is consistent with a temperature increase of 1 °C[50]. Because body temperature in birds fluctuates up to 5 °C with activity level[51], we speculate that the decreased motif duration resulted from the visibly increased activity when males were allowed to sing again. We thus do not attribute reduced motif duration to changes in syringeal muscle speed, which is also two orders of magnitude faster. The apparent increased motivation to sing may have driven reduced motif duration by increasing circuit temperature. (*iii*) Source level. Our data shows that singing prevention caused a reduction in motif source level, which cannot be explained by an increased motivation that typically correlates with an increase in source levels[52]. Instead, we propose source level reduction is caused by reduced subsyringeal pressure due to weaker abdominal muscles. Unfortunately, we did not measure muscle properties of these muscle groups and this hypothesis requires testing. Taken together, based on current mechanistic insights on vocal production, we propose that motif duration changes are caused by increased temperature of neural circuits, and we attribute changes in

$f_o$, WE and source level specifically to exercise-induced changes in syringeal and respiratory muscle physiology.

Our data strongly suggest that song exercise is a previously unrecognized cost of adult song maintenance. Such costs are expected for sexually selected signals, but in birdsong direct metabolic costs of singing are low[53–55]. Consequently, with the low physiological costs to produce song, costs were considered mostly developmental and song an honest indicator of past condition (the developmental stress hypothesis[56]). Our data suggest that exercise time needed for maintaining peak performance is a previously overlooked cost to adult singing. Indeed, supplementing food to birds in the wild increases their song production[57,58] suggesting a trade-off between foraging and singing. Many songbird species sing daily - in a dawn chorus or alone - to tie and maintain social bonds as well as defend territories[2], but surprisingly also commonly outside of these contexts[59]. Zebra finches sing hundreds of songs per day even when being isolated from conspecifics in captivity[18,19], and in the wild they keep singing daily even under extremely adverse conditions[20]. The need for daily exercise to maintain peak performance provides a mechanistically powerful explanation for birds to sing daily.

Our data also provides an intriguing explanation of the sexual dimorphism of the syrinx in zebra finches[60]. In zebra finches, the male and female syrinx is indistinguishable until the onset of song learning in males around 20 days posthatch. After this point, the male syrinx muscles gradually increases in mass[16] and contraction speed[14,15] and decreases in MIS (Fig. 1b), while female syrinxes stay similar to the juvenile state. This has previously led us to propose that sex differences are primarily driven by use and secondarily set by hormonal influences[17]. Our current data supports this hypothesis, as the influence of hormones was not sufficient to develop and maintain normal male syringeal muscle morphology, gene expression and physiological performance. As bone remodels by the forces exerted on it[61], the sexually dimorphic appearance of the zebra finch syrinx, may be driven largely by use and thus by the sexually dimorphic brain. This finding thus opens fascinating questions on how female syrinxes develop postnatally in species where both sexes sing[62–64]. Such species will provide unique opportunities to study the interactions between exercise and hormones on postnatal development and vocal behaviour.

Our data imply that the neural regulation of fibre type plasticity of syringeal and laryngeal muscles shares similarities, and differs from limb muscles. In well-studied limb muscles, loading paradigms increase CSA and mitochondrial function, and generally transform fast into slower fibre types[12,65], while unloading paradigms have the opposite effect, decreasing CSA and mitochondrial function and transforming slow into faster, more fatigable, fibre types (Supplementary Fig. 3). Our data establish that syringeal muscle fibres also exhibit plasticity of CSA, mitochondrial function, and fibre type by neural regulation and the direction of CSA and mitochondrial function plasticity is consistent with limb muscle. However, in terms of muscle speed, syringeal muscle fibre types respond oppositely to limb muscle: loading increases speed. This opposite response is also observed in laryngeal muscle (Supplementary Table 1). The physiological, anatomical and proteomic changes induced by our singing prevention paradigm are less strong but of same direction as changes induced by denervation, which supports the idea that both represent unloading paradigms of different strength. Together, the currently available data suggests a unified regulatory model for syringeal and laryngeal muscles where unloading (disuse, denervation) transforms vocal muscles from superfast to fast to slow fibre types, and vice versa in response to stimulation (song, electrical stimulation) (Supplementary Fig. 3). Whether the opposite polarity of speed control in vocal muscles is due to neurogenic or myogenic factors remains unknown, but we expect that myogenic ontogeny plays a major role. Embryonic development places most laryngeal muscles in the craniofacial muscle group[13]. The expression of *MYH13*[14] and earlier observations[66] place syringeal muscle also in the craniofacial muscle group, but the embryonic development of syringeal tissues needs further study to establish myoblast origins. The molecular mechanisms underlying fibre type regulation in limb muscles have been partially uncovered[11], but require investigation in laryngeal and syringeal muscles. To further test our proposed regulatory model we suggest loading mechanism can be best studied after experimental or natural periods of unloading, e.g., in seasonally singing birds, because normal healthy syringeal muscles seem to have reached the maximum speed possible for vertebrate synchronous muscles and are thus maximally loaded[14].

Experimental challenges in studying human laryngeal muscular function in vivo, has heeded the explicit call for animal models that can be trained in paradigms similar to human vocal training[9]. Songbirds provide unique opportunities because (*i*) their song is a learned, stereotyped behaviour that is produced like human speech[5,46], unlike rodents[67], (*ii*) the song motor system -from neural circuits to the vocal organ- is dedicated to song production and discrete from other vital functions and accessible to experimental manipulation[17], which (*iii*) enables acquisition of longitudinal datasets[19]. Songbirds furthermore (*iv*) can be trained on complex combinatorial targets for all motor systems[68], to test coordination between motor systems which is a major pedagogical training goal of human vocal exercise[9], (*v*) and provide molecular targets to model e.g. Parkinson's disease[69]. (*vi*) We show that syringeal muscles respond strongly to use changes, and in short time frames compared to laryngeal muscles. Therefore, we propose songbirds as a powerful animal model to study pathophysiology of human voice disorders concerning laryngeal muscle function and for further development of therapeutic interventions and pedagogical tools for voice therapy and (re)habilitation of laryngeal muscles.

## Methods

### Animals

Subjects were 79 male and 13 female lab-bred zebra finches (*Taeniopygia castanotis*). Animals ($N = 79$ males) at the University of Southern Denmark (SDU) were group housed in 3x3x2 m aviaries on a 12:12 h light:dark-cycle and provided with food, cuttlefish bone and water ad libitum. All experiments and procedures were performed in accordance with the Danish Animal Experiments Inspectorate Copenhagen, Denmark (2019-15-0201-00308).

Female preference tests were conducted at Leiden University, the Netherlands. Birds ($N = 13$ females) were housed in $175 \times 80 \times 200$ cm aviaries on a 13:11 h light:dark-cycle and provided with food, cuttlefish bone and water ad libitum. At least one week prior to testing animals were moved in groups of 2-4 into smaller holding cages. Experiments were approved by the committee for animal experimentation at Leiden University and the Centrale Commissie Dierproeven (CCD) of the Netherlands (1160020186606) and monitored by the Animal Welfare Body of Leiden University, in accordance with national and European legislation.

### Muscle use intervention paradigms

**Nerve cuts.** We prevented use of syringeal muscles by unilateral section of the hypoglossal nerve. Surgery was performed under a dissecting microscope. Inhalation anaesthesia was induced (3%) and maintained (1%) with isoflurane (Baxter). A 10 mm lateral incision was made in the skin of the neck exposing the trachea with the bilateral tracheosyringeal branches of the hypoglossal nerve (NXIIts). We removed at least 5 mm of the right branch by cutting the nerve with a pair of micro-scissors from about 10 to 15 mm cranial from the syrinx. The skin incision was closed with 8/0 unifilament suture (AroSurgery). Following surgery, the animals recovered in a 20x10x10 cm cage in a heated chamber (39 °C), and returned to their recording box. Nerve cuts were either performed in juvenile males before the start of song learning (30 ± 1DPH) or in adult males (>100DPH). The juveniles were sacrificed at 65- and adults at 2- and 21-days post-surgery. Moreover, successful long-term denervation

was supported by the disappearance of axon-related proteins in the protein profiling experiments as well as immunohistochemistry for neurofilament (see below). During dissection we carefully checked for signs of reinnervation, but didn't find evidence in any animal.

**Singing prevention.** To study the effects of a much milder muscle unloading paradigm that is also more physiological and behaviourally relevant compared to nerve cuts, we prevented males from singing. Adult males were kept in custom-built, sound attenuated recording boxes (60x95x57 cm) and vocalizations were recorded continuously[70]. Males were moved to recording boxes and monitored several days before entering the singing prevention paradigm to ensure they were habituated to the new environment.

Previous work showed that preventing birds from singing by keeping them in the dark for 5 h did not elevate blood corticosterone levels compared to control conditions[40]. However hanging up to 1.5x body weight on their neck induced a postural change which prevented song, but also increased corticosterone levels and reduced body weight, and thus increased physiological stress[42]. Therefore, we preferred a darkness paradigm.

To prevent males from singing they were kept in the dark for 7 consecutive days, except for two to four 30-minute long feeding sessions spaced evenly throughout a day. During the feeding sessions, males were allowed to call, eat and drink ad libitum, but interrupted from singing attempts by distracting the animals. Bodyweight was monitored throughout the experiment to guarantee the animals' wellbeing and did not change during the experiment. All vocalizations were monitored continuously to ensure that birds wouldn't start singing in the dark. On the morning of day 8 the original colony light schedule was resumed, and males could sing undisturbed.

## Muscle physiology

Muscle fibre bundles were prepared and stimulated in vitro to record isometric force responses as previously described[71]. In brief, birds were sacrificed by an isoflurane overdose and the syrinx was exposed, isolated and pinned down on Sylgard-covered Petri dishes in cold, oxygenated dissection buffer[72]. Fibre bundles were obtained from the *m. tracheobronchialis dorsalis* (DTB). Muscle fibre bundles consisted of a subsection or the entire (denervated as juveniles and juveniles) DTB. Muscle preparations were mounted in a temperature-controlled bath perfused with oxygenated recording buffer[72]. The rostral end of the preparation was fixed to a force transducer (Model 400 A, Aurora Scientific) and the caudal end to a micromanipulator to control preparation length. After mounting, the muscle preparation was allowed to rest for 20 minutes. Muscle fibres were stimulated by a high-power follow stimulator (Model 701 C; Aurora Scientific) at constant voltage using platinum electrodes. Force and stimulation signals were low-pass filtered at 10 kHz (EF120 BNC, Thor Labs) and digitized at 20 or 40 kHz (PCI-MIO-16E4, National Instruments). After optimizing the stimulation parameters at $39.0 \pm 0.1 \,°C$ (as detailed in[15]), we obtained seven twitch contractions and three 100 ms duration tetanic contractions at optimal length and stimulation frequency. All software to control the setup and record data was written in Matlab (MathWorks, RRID:SCR_001622). As a measure of contraction speed, we extracted the full width at half maximal (FWHM) force of single twitch stimulations, which is the time from crossing 50% force increase to decrease, and equals the previously reported $t_{50-50}$[71,73]. Maximal isometric stress was calculated as maximal tetanic force divide by the muscle cross-sectional area (CSA) estimated from optimal (resting) length $L_0$, the dry weight (dry-wet conversion factor: 5) and density ($1060 \,kg/m^3$ from[74]) of the muscle fibres.

## Muscle morphology

Birds were sacrificed by an isoflurane overdose and immediately perfused transcardially with dissection buffer[72] to remove blood cells before dissecting the syrinx on ice at SDU. Syrinxes were dried on

kimwipes and frozen in an isopentane bath cooled by liquid nitrogen, and subsequently stored in cryotubes (Nunc X) at -80 °C. All samples were shipped on dry-ice to Umea University for sectioning and staining.

**Immunohistochemistry.** Serial muscle cross-sections, 8–10 µm thick, were cut in a cryostat (Leica CM3050S cryostat, Leica Biosystems, Nussloch Germany) at −22 °C and mounted on Superfrost Plus Adhesion Microscope Slides (Menzel Gläser, Menzel GmbH & Co). We used cross-sections of the adult male zebra finch syrinx at midbody (between the most rostral part of bronchial rings B2 and 3) where all muscle fibres are present. Immunohistochemical staining was performed using well-characterized antibodies and modified standard immunohistochemical techniques. For antibody (AB) specificity and concentrations, see Table S2. In brief, the sections were immersed in 5% normal non-immune goat serum (DakoX0907, Agilent Technologies Inc., CA, USA) for 15 min and thereafter rinsed in 0.01 M phosphate-buffered saline (PBS) for 3x5 min. The sections were then incubated with the primary ABs diluted to appropriate concentrations in PBS with bovine serum albumin in a humid environment. Incubation was carried out overnight at 4 °C for ABs M4276 (1:500) and CBL212 (1:500) and the next day the sections were double stained with polyclonal AB L9393 (1:500), for 1 h at 37 °C. After additional washes in PBS, the sections were incubated with the secondary ABs (37 °C for 30 min) and washed in PBS 3x5min. Bound primary ABs were visualized by indirect immunofluorescence using corresponding secondary ABs conjugated with fluorochromes with different emission spectra; Goat anti-Mouse IgG Alexa Fluor 488 (A-11029, 1:1000, Invitrogen by Thermo Fisher Scientific, Rockford, USA) and Goat anti-Rabbit IgG Alexa Fluor 568 (A-11036, 1:1000, Invitrogen by Thermo Fisher Scientific, Rockford, USA). The sections were thereafter washed in PBS for 3x5 min and then mounted in ProLong Gold antifade mountant or ProLong Gold antifade mountant with DAPI (4′,6-diamidino-2-phenylindole), for staining the nuclei (P36930, Invitrogen by Thermo Fisher Scientific, Life Technologies, Oregon, USA).

**Morphometric analysis.** All immunohistochemically stained cross-sections were captured using a digital camera (Leica DFC360 FX) connected to a fluorescence microscope (Leica DM6000B, Leica Microsystems CMS GmbH, Wetzlar, Germany) with a motorized table. Individual images were captured across the whole section with a 10x magnification objective to generate a high-resolution montage of the whole section. The resulting images were around 20,000 ×30,000 pixels. To separate left from right muscles, we made binary masks in Photoshop identifying left and right pixels. To significantly reduce computational time, we rescaled images and masks to 10,000 pixels height.

We detected all muscle fibres per side using an automated procedure based on the laminin-stained cell borders. Analysis was implemented in Matlab (MathWorks, RRID:SCR_001622). We improved contrast of the laminin layer using contrast-limited adaptive histogram equalization (*adapthisteq* function) and applied the mask. We then converted the image layer first to grayscale using a global threshold[75] and second to a binary image with a 10-20% reduced shift to detect all cell borders. We removed noise areas and small negative areas (*bwareaopen* function) that were blood vessels, nerves and SR. This resulted in detection of areas between laminin containing myofibrils. To smoothen the detected myofibrillar area per muscle fibre we applied a diamond-shaped morphological structuring element to subsequently grow and shrink the areas (*imdilate* and *imerode* functions). To isolate individual muscle fibres in the binary image, we computed the distance from border to center for all areas (*bwdist* function), suppressed local minima using the H-minima transform[76] (*imhmin* function) and computed the watershed segmentation (*watershed* function). The segmentation accurately detected the

myofibrillar area of the muscles, but thus left out a large area in between fibres containing other cellular components. To estimate the total muscle CSA per hemisyrinx we therefore dilated all fibres with a diamond-shaped morphological structuring element (radius 10) to fill up the space in-between fibres.

For each resulting fibre area, we computed multiple morphological features (area, perimeter, eccentricity, solidity, major and minor ellipse length and ratio, circularity) and used feature constraints to remove false detections. We used the detected areas to measure mean MY-32 expression intensity per fibre on the MY-32 image layer. To quantify the fibre area and MY-32 expression distribution we calculated histograms with fixed bin widths. To quantify changes in these distributions due to experimental interventions, we calculated probability density function estimates to correct for differences in the total number of fibres in each dataset (*histcounts* function, '*pdf*' normalization). We considered fast fibres to have a mean MY-32 expression intensity >50, and superfast muscle (SFM) fibres a mean MY-32 expression intensity ≤50. This boundary clearly separated the unstained from stained fibres in the untreated condition. The obtained fractions of 66% SFM fibres correspond well with the 67% extracted from Mead et al. [14], who used a Kolmogorov-Smirnov optimization to separate two populations of fibres in a single syringeal muscle (DTB), and are lower than the 87% reported in Christensen et al. [22], who manually counted fibres of the entire syrinx.

## Proteomics

To determine the relative abundance of muscle proteins in the DTB, 0.6-1.2 mg samples were shipped on dry ice to the University of Vermont and analysed by label-free proteomic analyses as previously described[77]. Briefly, the muscle samples were solubilized in Rapigest SF Surfactant (Waters), reduced, alkylated, and digested with trypsin (Promega). The resulting peptides were separated by ultra-high pressure liquid chromatography and directly infused into a Q Exactive Hybrid Quadrupole-Orbitrap Mass Spectrometer (Thermo Fisher Scientific). Data were collected in data dependent mode and recorded in .raw files. Peptides were identified and liquid chromatography (LC) peak areas were determined using Proteome Discoverer 2.2 to search against the zebra finch database downloaded from UniProt (11/22/2022, [https://www.uniprot.org/taxonomy/59729]). The searches accounted for the presence of the following posttranslational modifications: oxidation (M, P), oxidation (M), phosphorylation (S, T, Y), carbamidomethyl (C), and acetylation (N-terminus of protein). The LC peak areas were exported into Excel and the abundance of the top 3 ionizing peptides from each protein isoform of interest were used to quantify protein abundances. Results were manually curated to remove redundant peaks and gene names were curated to follow the HGNC nomenclature[78]. After removing redundant entries, LC peaks were normalized to the total sum of peaks and Histone H4 (Uniprot accession: B5FXC8, [https://www.uniprot.org/uniprotkb/B5FXC8/entry]) (Supplementary Data 2). Expression of all MyHC genes belonging to the fast/developmental cluster on chromosome 18 (NC_044230.2 5,745,795-6,139,115, [https://ncbi.nlm.nih.gov/nuccore/NC_044230.2/]), except MYH13 were quantified together using LC peaks from shared peptides and the protein is called MYH-fast in the entire MS. This strategy was chosen as not all gene models were identifiable by unique peptides. Proteins were grouped into 4 groups: sarcomeric, Ca-handling, mitochondria and "other" based on their subcellular localization and function. Information on subcellular localization and or function of proteins was extracted from Uniprot (https://www.uniprot.org/), gene cards (https://www.genecards.org/) and literature searches.

To correct for contamination with proteins expressed in red blood cells, LC peaks originating from proteins known to be highly abundant in red blood cells[79] were excluded from the analysis (Uniprot IDs A0A674HE74, B5FXM1, B5G3P7, H0ZSY4, A0A674HG19). Muscle samples from animals in the singing prevention paradigm were not transcardially perfused prior to tissue sampling because short dissection times were prioritized to perform muscle physiological measurements in the same animals. These samples thus contain a higher amount of blood proteins than the samples from animals in the long-term denervation group. Because avian erythrocytes are nucleated and we normalize protein expression to Histone 4, we might overestimate the effect of singing prevention on protein abundance.

## Song recording and analysis

Vocal output is the result of interactions between neural circuitry (aka song system) and motor systems[80]. Adult zebra finches sing an individual motif consisting of several syllables. Individual syllable production and sequence is highly stereotyped. Like human speech auditory feedback corrects the motor code for deviations from the song template[5,81]. Altered acoustic feedback can drive changes in e.g. syllable pitch over days[81], with first indications of such song system-induced corrections occurring the earliest after 6-12 hours[81]. We analysed changes in vocal output immediately (0-2 hours) after release from singing prevention where motifs are not yet compensated by error-correction of the song production circuitry and reflect changes in the vocal periphery.

In each recording box, sound was monitored continuously by Sound Analysis Pro[70] and recorded by an omnidirectional microphone (Behringer ECM8000) mounted 12 cm above the cage, digitized at 16-bit and 44.1 kHz (Roland octa capture, amplification 40 dB). Recording chain sensitivity was calibrated with a 1 Pa, 1 kHz tone (sound calibrator model 42AB, G.R.A.S., Denmark).

Per animal (N = 10 males) we defined a sound segment containing the motif and used cross-correlation to detect and isolate motif segments in all pre and post song files. We recorded 976±455 (median: 881, range: 554-2087) and 535±288 (median: 581, range: 82–934) motifs pre and post singing prevention respectively. We bandpass filtered the sound between 200-12,000 Hz using a $2^{nd}$ order Butterworth filter with zero-phase shift implementation (*filtfilt* function). We divided the signal into 4 ms duration bins with 0.5 ms steps and calculated the following acoustic features per bin: aperiodicity, power, source level, and $f_o$ using the Yin algorithm[82]. We aligned the motifs to the highest correlation of the source level (*finddelay* and *circshift* function) for the pre and post singing prevention separately. To ensure we compared acoustic feature trajectories of motifs with the same syntax, we focused on the most common syntax and omitted all other motifs.

We used a fixed source level (SL) threshold per individual to segment sound and silence within the motif, and removed segments of sound and silence below 30 ms and 40 ms duration, respectively. The remaining binary signal represented the syllables and was used to extract syllable on- and offset timing, syllable and gap duration. Motif duration was the summed duration of all detected syllables and gaps.

Next, we calculated the mean, minimal, maximal and range of $f_o$ per syllable, after removing spurious $f_o$ detections within syllables by removing jumps between adjacent bins over 100 Hz and $f_o$ trajectories below a duration of 5 ms. We omitted syllables where pitch detection was not robust. In total, we analysed 485±261 (median: 393, range: 233-1093) and 296±157 (median: 249, range: 81–532) motif iterations pre and post singing prevention.

## Female choice experiment

For the song preference test, we used a previously validated operant paradigm[27]. Nine female zebra finches were moved into one of nine identical experimental setups consisting of a wire-mesh cage (70 × 30 x 45 cm) with a solid back panel with operant keys, each placed in separate sound-attenuated chamber (height: 250 cm, width × length irregular quadrilateral minimally 106×158×304×387 cm). From the first (left) and fifth (right) of five equally spaced perches, a bird could peck a 5 cm diameter white piezoelectric circular plate (response key) with a

small embedded red light-emitting diode LED (5 mm diameter) at the top. When pecked a custom-built minicomputer (sound chip Oki MSM6388, Tokyo, Japan) and laptop (Sony Vaio E series, Sony, Minato) placed outside the experimental chamber registered the activation time of the response key and triggered an acoustic playback of an assigned stimulus via a loudspeaker (Vifa 10BGS119/8, Viborg) suspended from the ceiling at 1 m above the cage. Stimuli were played back at 70 dB re 20 µPa at the central perch (Voltcraft sl451 sound level meter, fast response setting, A-weighting).

**Motif selection and stimulus construction.** Because both total number of syllables, duration and source level are known to influence preference[28,83] we ensured that stimuli pre and post were equally long and loud. From all pre and post recorded songs, we first identified motifs that were within 4 ms duration and 1-2 dB SL of the mean motif duration and mean SL pre and post singing prevention (Fig. 3e). In one individual the pre and post SL differed so much we used a 30 ms and 6 dB range to find overlap. From the overlapping motifs, we randomly picked one pre and one post motif per male and used this to construct a natural song bout for each male. We exchanged the motifs in a bout sung by that male before singing prevention to retain a natural bout organization including introductory notes.

**Testing paradigm.** During all training and testing, experimenters were blinded with respect to whether a stimulus was recorded pre- or post-treatment. For training, females were first left to explore the cage with the operant set up on active (red LEDs and playback reward switched on during lights on) as some females quickly discover the link between key pecking and song reward by autoshaping. Females that did not start key pecking spontaneously within the first day were given training sessions twice daily for 20 min until operant responses were logged. During shaping, the experimenters (K.R., I.A., C.P.H.E., outside the test chamber behind a one-way mirror) initially drew females' attention to the keys by flashing the LED lights before playing the song reward and then gradually rewarding all behaviour leading to closer approach and exploration of the keys, taking care to reinforce the keys on both sides (details on training, see[31]). During initial training (involving 10 females and 10 stimulus sets), songs were randomly chosen from the pool of pre and post singing prevention to see whether females were motivated to peck for either stimulus category. During this first testing round 7/10 females learned to peck the keys and were highly motivated to hear songs ( > 30 key pecks/day) and as previously reported for this species females preferred longer songs[29,31]. For the second, actual preference test all successfully trained females plus 3 additional females were tested with the actual test songs.

**Preference analysis.** The pecking events and cumulative learning curves of each female were checked daily and preference testing began the day after the initial pecking had changed from incidental pecking to an exponential increase of frequent operant responses on both sides[31]. Each test lasted 4 days and each night we switched the assignment of the stimulus songs between left and right keys to control for side preferences. We calculated female preference as the sum of keypecks for one stimulus over 4 days divided by the sum of total keypecks. Individual female preferences were analysed with G tests against 50% chance level using Williams' correction.

**Housing details.** Birds in Leiden, the Netherlands, were housed and tested in climate regulated rooms (19–22 °C and 40–60% humidity) of the bird facility. Lights were on from 0700 to 2030 h (starting and ending with a 15 min twilight phase). Seed mix was (Deli Nature 56-Foreign finches super, Schoten, Belgium) enriched with minerals and vitamins (GistoCal, Raalte, the Netherlands) and supplemented once per week with egg food. The majority of females had breeding experience, and all had been housed with males at least in hearing distance.

## Statistics
No formal methods to predetermine sample size were used; sample sizes are similar to those used in the field. Randomization and blinding was performed when analysis wasn't automated and details are described in each section of the methods. Statistical results and setting for all tests applied are presented in Supplementary Data 1. Data are presented as mean values ± 1 S.D throughout the manuscript.

## Reporting summary
Further information on research design is available in the Nature Portfolio Reporting Summary linked to this article.

## Data availability
The proteomic data generated in this study have been deposited in the MassIVE database (Dataset: MSV000091352) (https://massive.ucsd.edu/ProteoSAFe/dataset.jsp?task=1a7f2f007a3247bf998adde62872cd2e). The processed proteomics data are available as Supplementary Data 2. Detailed statistical results are provided in Supplementary Data 1. Source data are provided as a Source Data file. All other data are available from the corresponding authors upon reasonable request. Source data are provided with this paper.

## Code availability
Code is available on reasonable request from authors.

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

## Acknowledgements

Peter Snelderwaard, Maria Anthonsen, Bianca Jorgensen and Anna-Karin Olofsson for technical support. Sonja Jacobsen, Emilie Radoor and Emilie Jensen for animal care. This work was supported by Carlsberg Foundation CF17_0949 (IA), Villum Fonden 36004 (IA), National Institutes of Health grant NIH R01 HL157487 (MJP), National Institutes of Health grant NIH R01 NS084844 (CPHE), NovoNordisk grant NFF20OC0063964 (CPHE).

## Author contributions

I.A. and C.P.H.E. designed experiments. I.A. carried out the muscle physiology experiments and analyses. P.S. carried out the fibre typing experiments, C.P.H.E. carried out the image analyses. N.W., M.J.P. conducted the proteomic data and M.J.P. and I.A. analysed the proteomics dataset. I.A. carried out the acoustic recordings and C.P.H.E. analysed the data. I.A., K.R and C.P.H.E. conducted the female choice experiments and I.A. carried out the analysis. I.A. and C.P.H.E. wrote the first draft of the manuscript, all authors contributed to the final draft.

## Competing interests

The authors declare no competing interests.
