## [Peer Review File · Nature Communications]

Daily vocal exercise is necessary for peak performance singing in a songbirdREVIEWER COMMENTS

Reviewer #1 (Remarks to the Author):

The authors present interesting data in this manuscript. On the one hand, they show how unilateral denervation changes composition of the syringeal muscles. On the other hand, they prevent zebra finches from singing and show data that indicate that songs after this deprivation are less preferred by females. From these disjunct data sets, the important conclusion is reached that daily singing is required for peak performance. Unfortunately, the two stories do not represent a coherent set of data. Furthermore, a number of interpretations are highly speculative and do not comprehensively account for the large body of knowledge on zebra finch vocal behavior and its neural control. These general points are discussed below.

Major Points:

1. Denervation of the syringeal muscles does not represent 'disuse' in the sense of not singing daily. It causes very different changes in muscle than disuse (as for example in the case of immobilization of a leg in a cast). Denervation of muscles is known to alter their phenotype much more drastically than simple disuse. It is known from other systems that fiber type composition changes much more rapidly than is the case with disuse. In other words, the form of atrophy is not comparable between the two manipulations. This problem leaves the impression that in the two parts of the manuscript apples and oranges are compared. Why has the muscle proteomics work not been done on syringeal muscles from males that were not allowed to sing?
2. The behavioral experiment, where males were prevented from singing by being kept in the dark lacks important controls. How do we know that the documented changes in song are not due to other effects? For example, being kept in the dark may have changed the motivation to sing. We know that motivational aspects change motif duration (e.g., directed versus undirected singing). Furthermore, being kept in the dark must have altered melatonin production, and we know that melatonin affects song tempo. Thus, the changes may have nothing to do with disuse of the syringeal muscles.
3. Interpretations that are questionable are detailed below. Additionally, there is very selective and biased use of the vast literature.

Specific comments:

- l. 20: In light of the above criticism, this conclusion is not supported.
- l. 36: Here the premise of the paper is stated. Neither part of the manuscript really tests this question directly and unambiguously
- l. 55: Descriptions of changes after denervation merely confirm what has been found earlier (Julie Wade's work).
- l. 74: Again, these data do not show that daily exercise is required. They show that denervation leads to the documented changes – but that is very different from how daily exercise may maintain muscle phenotype.
- l. 189: This conclusion is not merited from the presented data (see major points).
- l. 196: Does the decrease in frequency correspond to the decrease in amplitude?
- l. 200: Again, there are other possible explanations for the observed changes. This conclusion is

therefore not unequivocally tested.

I. 203: Yes, but more recent work directly on zebra finches (and not Bengalese finches) has added to our knowledge.

I. 206: If muscle speed and motif duration decrease, how is this an indication that muscle speed is responsible for the observed changes in song? This conclusion is not valid.

I. 211: The temperature connection is only one possible interpretation. Melatonin may play a part, as does motivation. Again, this is a very selective use of different aspects known to affect motif duration.

Discussion: Because of the many concerns with the data sets and their interpretations, most of the points raised in the discussion are less clear than stated. This needs to be toned down in light of the many reservations about the disjunct parts.

Reviewer #2 (Remarks to the Author):

Peak performance singing requires daily vocal exercise in songbirds.

This is a very neat discovery and a beautifully executed set of experiments. I have no doubt this will be a much-cited, significant contribution to the literature in this field, as well as broadly relevant (and interesting) outside of the songbird or vocal production world. I do not have major criticisms of the science, but only a few minor comments and suggestions.

Generally, the manuscript is extremely dense, which is understandable given the amount of work presented here, but in the relatively short format for this journal, it makes clarity a bit of an issue.

Abstract

Aware of the very short format of abstracts for the journal, but the result is a few sentences which are not clear:

line 17 "...it is unknown whether vocal, like limb muscles, exhibit..." ideally this should read "vocal muscles, like limb muscles" as the reader may not be aware that there even are vocal muscles. I had to read it a couple of times to clock that "vocal" modified "muscles", and rather thought that the authors had just left off a word somewhere.

Line 19 Similar issue with "Preventing singing rapidly alters" where it is not clear if the authors mean that the prevention of rapid singing alters muscles, or if the prevention of singing results in rapid alteration of muscles.

Results

Line 47, not sure about use of "vocal folds" here as the vibratory structures in the oscine syrinx are not really vocal folds (despite similar function).

Line 53, would be very helpful to give a functional definition of maximal isometric stress here. I think for the lay-biologist it is clear what twitch contraction and contraction speed would mean both in terms of

muscle behaviour, but MIS is more oblique and its relevance may be lost for much of the intended readership (especially considering how central this change is to your main findings).

This is also so for the legend of Fig 1, where the functional definition of FWHM is not provided until deep in the methods at the end of the paper. I think readers that are not muscle physiologists will struggle to put together the significance of these measures as the manuscript is currently presented.

Line 195, were these changes statistically significant between pre- and post- treatment birds? For example, source level difference does not seem particularly convincing given the size of the errors? Edit- I see some p-values in Fig 3 to support this. Regardless of statistical significance, the behaviour experiment here would suggest that these changes are biologically relevant, I suppose. It is amazing that females seem to show such a strong preference for what seem to me to be relatively minor changes. Do the authors have ideas about which of these changes seem more salient to females?

What is the point of the “duration decrease” bar in Figure 2, panel F? Isn't this information already in the left graph of panel 3F? is a graph really necessary there? Perhaps this needs a clearer explanation.

Figure 3, panel D, again this just needs a bit more description. What goes into the colour mapping of each row in the aligned motifs? Is this a measure of consistency across multiple repeats of the motif in each condition?

Discussion

I am curious as to why calling would not be adequate to exercise syringeal muscles, and what this means for female syringeal development and maintenance?

Do the syrinxes of females (in species without female song) remain in a paedomorphic condition? Or is there a maturation, and would a female prevented from calling be expected to lose muscle mass and contractile speed?

Methods

Line 347, please include a note of how far cranially/caudally the section of NXIIIts was removed (or did this vary between individuals?)

Line 302-4, this is an interesting idea, but it is not clear how the results presented here actually support it (other than contraction speed decreases with disuse), is the only difference between superfast and fast twitch fibres simple a matter of usage? Could the authors provide a bit more clarity in what the hypothesized mechanism would be here?

Line 312, is there a missing (or redundant) word here: “A complete lack of both access and experimental access to human laryngeal...”?

Line 321, I agree that these results are really broadly relevant and can aid in better understanding motor plasticity in response to training. Nonetheless the differences in the tissue composition of mammalian vocal folds and syringeal vibratory masses I think make this final claim a bit too bold, as mammalian vocal disorders are often linked to pathologies of the vocal folds themselves (which are themselves muscular and vascularised in a way that avian syringeal labia are not), rather than the laryngeal muscles

that control their position and tension.

Supplementary Text -

The authors state here that it is still not known how the two muscle fibre types differ in their responses to exercise. In the histological investigation of muscle cross sections, it is not possible to compare responses to denervation by superfast and normal fast twitch fibres, and compare for example relative decrease in diameter at the minimum?

General comment, please be consistent in either British spelling or American spelling (e.g. fiber and fibre both occur in current text)

Reviewer #3 (Remarks to the Author):

Summary -

I have reviewed the paper "peak performance singing requires daily vocal exercise in songbirds and found it to be technically sound to the best of my knowledge, but the presentation of the research question and the objectives and hypotheses are not clearly stated, nor is there a lay summary of the approach leading into the technical results section which would be helpful. The extremely brief introduction is inadequate in its goal of introducing the reader to the overall question, hypotheses, objective, and approach and therefore the rest of the manuscript is difficult to navigate and further the importance and implications of this carefully done work are not apparent. I recommend the authors make a more thorough attempt at a concise yet useful introduction.

Introduction: Even for a short format journal like Nature communications this introduction is inadequate. It fails to communicate the context of this study, the basis for the hypotheses, or the timelines of the approach.

Results:

The results are clearly presented, although the use of acronyms for a general audience journal is excessive. Consider limiting the number of acronyms when possible. The statistics section indicates that the details of the statistical tests are presented for each section of the results - this is not the case. Please clearly indicate the statistical approach used for each set of experiments somewhere accessible in the main manuscript.

Discussion: Overall the discussion is thoughtful but limited in scope. Can the authors expand the scope to include more ecological/evolutionary or bigger picture implications of these findings? Also more discussion on all the changes in proteins etc highlighted in fig 2 would be helpful. The data in fig 3 seems particularly interesting, but it is all but overlooked.

Overall the authors have conducted a technically sound study, but the manuscript does not do it justice, especially for a broad scientific audience.

Point-by-point response to reviewers NCOMMS-23-12907

We thank the reviewers for their effort to thoroughly and carefully review our paper, and for the positive and constructive comments on our manuscript. In the revised manuscript we have addressed the concerns that they have raised, as described below. We have included short sections of rephrased text in the rebuttal with new text underlined and refer to line numbers in the revised *track changes* document. To improve readability and avoid repetition, we did not include longer new sections in the rebuttal, but refer to line numbers only.

In our revised manuscript, we have significantly expanded both introduction and discussion, and added new proteomics data on the singing prevented birds (**Supplementary Fig S3**).

Reviewer #1

The authors present interesting data in this manuscript. On the one hand, they show how unilateral denervation changes composition of the syringeal muscles. On the other hand, they prevent zebra finches from singing and show data that indicate that songs after this deprivation are less preferred by females. From these disjunct data sets, the important conclusion is reached that daily singing is required for peak performance. Unfortunately, the two stories do not represent a coherent set of data. Furthermore, a number of interpretations are highly speculative and do not comprehensively account for the large body of knowledge on zebra finch vocal behavior and its neural control. These general points are discussed below.

Major Points:

Q1. Denervation of the syringeal muscles does not represent 'disuse' in the sense of not singing daily. It causes very different changes in muscle than disuse (as for example in the case of immobilization of a leg in a cast). Denervation of muscles is known to alter their phenotype much more drastically than simple disuse. It is known from other systems that fiber type composition changes much more rapidly than is the case with disuse. In other words, the form of atrophy is not comparable between the two manipulations. This problem leaves the impression that in the two parts of the manuscript apples and oranges are compared.

A1: We agree with the reviewer that different methods to abolish muscles activity have different effect strengths. Indeed, it is well established that denervation typically induces more drastic changes compared to e.g., leg or joint immobilization, hindlimb unloading, bed rest or spinal cord injury in animals including humans.

The current general consensus on the neural regulation of muscle fibre type in limb muscles is that manipulations can be divided into **loading** and **unloading** paradigms and muscle fibres shift type dependent on use (e.g. (Ohlendieck 2011)). Although the molecular and cellular mechanisms underlying and driving changes after different (un)loading paradigms may be different, the *direction* of the phenotypic changes is typically consistent within unloading versus loading paradigms. Thus, denervation indeed can cause more rapid and severe shifts to e.g. faster fibre phenotypes or reduced CSA (atrophy) compared to milder unloading paradigms such as bedrest, but the effect *direction* is the same. This is well documented in exercise physiology literature. We thus respectfully disagree with the reviewer that we are comparing apples and oranges; instead, we are using two unloading paradigms with different strength.

Our aim was to test the hypothesis that syringeal muscles exhibit plasticity. To test this hypothesis, we needed to show phenotypic plasticity and chose one of the strongest intervention paradigms

(i.e., denervation) to induce the largest effect. We did not aim to investigate the molecular mechanisms underlying different unloading paradigms causing the phenotypic changes. This is certainly very interesting, but only relevant once a phenotypic effect has been established, and outside the scope of this manuscript. Our data clearly show that denervation leads to drastic changes in muscle speed, MIS, fibre type composition, protein composition and CSA (**Figs 1, 2, S1**).

That is why in the next phase of our study, we used a much milder unloading paradigm which is more physiologically and behaviourally relevant than denervation, namely singing prevention. Our data show that our singing prevention paradigm also drives changes, and that these changes are in the same direction as denervation but of lesser magnitude: In adults, denervation induced changes in FWHM and MIS (**Fig 1E,F**) that are stronger but in same direction as singing prevention (**Fig 3B**). In our revision we also included proteomics data of the singing prevented animals (see next question **Q2** and **Figure S2**), and those data are also consistent with this idea; proteome changes are less strong but in same direction as after denervation.

Taken together, singing prevention drives changes in the same direction as denervation, and can be viewed as an unloading paradigm with different strength.

In our revised manuscript, we have completely rewritten our introduction to clarify the abovementioned issues (**Line 44-61**). Furthermore, we have added the following sentence in the results section regarding singing prevention:

Main text (Line 337-340): “Because the short-term singing prevention paradigm affects speed, MIS and proteome in the same direction as short-term denervation, it acts as a milder unloading paradigm compared to nerve cuts.”

And added the following section in the methods:

Methods (Line 744-745): “Singing prevention. To study the effects of a much milder muscle unloading paradigm that is also more physiological and behaviourally relevant compared to nerve cuts, we prevented males from singing.”

In our discussion we significantly rewrote the paragraph on neural regulation of muscle fibre plasticity (**Line 501-524**)

Q2. Why has the muscle proteomics work not been done on syringeal muscles from males that were not allowed to sing?

A2: We did also collect tissue of males that were not allowed to sing for proteomic analysis. On one syringe side we measured physiological performance, and the other side was kept for proteomic analysis. Because we prioritised the muscle physiology measurements, we prioritised minimizing the dissection times. Because of this choice, we dissected the tissues in these males without prior transcatheter perfusion with Ringer’s solution. Consequently, the proteomes of these tissues included higher amounts of blood proteins. As our original submission already contained many diverse datasets, we decided not to include this analysis.

In our revised manuscript we now also include the proteomic analysis of singing prevented males. To correct for the blood proteins, we excluded proteins that are highly expressed in erythrocytes. Because avian erythrocytes are nucleated and we used the commonly used nuclear protein (Histone 4) for normalization throughout the manuscript, we may overestimate the effect of singing prevention on protein abundance and state this specifically.

We added the following section to our methods:

Methods (Line 895-903): “To correct for contamination with proteins expressed in red blood cells, LC peaks originating from proteins known to be highly abundant in red blood

cells⁷⁵ were excluded from the analysis (Uniprot IDs A0A674HE74, B5FXM1, B5G3P, H0ZSY4, A0A674HG19). Muscle samples from animals in the singing prevention paradigm were not transcardially perfused prior to tissue sampling because short dissection times were prioritized to perform muscle physiological measurements in the same animals. These samples thus contain a higher amount of blood proteins than the samples from animals in the long-term denervation group. Because avian erythrocytes are nucleated and we normalize protein expression to Histone 4, we might overestimate the effect of singing prevention on protein abundance.”

We observe that most proteins are downregulated and in between wildtype and long-term (21 days) denervation. The effects are thus in the same direction as after long-term denervation, but weaker. The results from the proteomics parallel our muscle physiology results from these animals, where contraction speed decreased in animals prevented from singing, but not as strongly as after 3 weeks of denervation.

We added the data to the Supplementary Information as **Supplementary Figure 2** and refer to it in the text:

Main text (Line 334-337): “Additionally, we quantified changes in protein abundance and composition and found that protein abundance of sarcomeric, mitochondrial as well as calcium handling proteins was reduced, while the composition of the MyHC-pool was less affected (Supplementary Fig 2).”

Q3. The behavioral experiment, where males were prevented from singing by being kept in the dark lacks important controls. How do we know that the documented changes in song are not due to other effects? For example, being kept in the dark may have changed the motivation to sing. We know that motivational aspects change motif duration (e.g., directed versus undirected singing). Furthermore, being kept in the dark must have altered melatonin production, and we know that melatonin affects song tempo. Thus, the changes may have nothing to do with disuse of the syringeal muscles.

A3: We agree with the reviewer that our experimental paradigm of keeping birds under extended dark conditions likely could have consequences for physiological parameters independent of singing activity. We absolutely considered these in our experimental design, and apologize that our discussion of the effects of our singing prevention paradigm was incomplete.

The change in lighting conditions is likely to affect A) melatonin signalling, B) has the potential to induce stress and C) may alter the motivation to sing. All three factors can affect the acoustics of animal vocalization, but none of them in the direction we observe here. Furthermore, these factors especially affect song duration and source level, which are the acoustic features that we i) do not attribute to changes in syringeal muscle physiology and ii) were excluded as cues in our female choice experiments.

In support of our conclusion, a recent study from the Koijma lab (Daisuke et al. 2023) that uses a very different singing prevention paradigm, reports changes in acoustics consistent with ours.

In conclusion, we think that some of the observed changes in vocal output (f_0 , f_0 range and WE changes) can be attributed to changes in syringeal muscle physiology as measured in the same individuals.

In our revised manuscript, we added an entire new paragraph discussing melatonin and other potential physiological factors (**Line 480-497**). Furthermore, we moved the paragraph discussing the causes of song output change from the results to the discussion and now specifically spell out our thoughts on how changes in muscle physiology have driven the observed changes in vocal output, based on current mechanistic insights on vocal production (**Line 498-523**).

Furthermore, we added the following sentence in the methods:

Methods (Line 749-753): “Previous work showed that preventing birds from singing by keeping them in the dark for 5h did not elevate blood corticosterone levels compared to control conditions⁴⁰. However, hanging up to 1.5x body weight on their neck induced a postural change which prevented song, but also increased corticosterone levels and reduced body weight, and thus increased physiological stress⁴². Therefore, we preferred the darkness paradigm.”

Q4. Interpretations that are questionable are detailed below. Additionally, there is very selective and biased use of the vast literature.

A4: It was not our intention to leave out any specific part of the literature. Our paper covers a diverse range of topics including muscle physiology, behaviour, behavioural ecology, motor control, and gene regulation. This is indeed a vast literature considering the reference limit for this journal, which is 70 references. As such we cannot avoid to sometimes cite reviews, instead of original research papers, while sometimes we need to cite very specific papers. We apologize if any bias arose, and we have tried to increase the spread of the literature in the revision. Without specific pointers we are not sure which part of the literature was uncovered, but we hope that with the significant changes in the introduction and discussion, the breadth of the literature cover has sufficiently increased.

Specific comments:

I. 20: In light of the above criticism, this conclusion is not supported.

This comment refers to the statement: *“Vocal output thus contains information on recent exercise status”*.

Thank you for this remark, which made us realise that in we might have been too brief in explaining this. We have now reworded several passages to explain in more detail how we reached this conclusion. Let us start with an analogy: The mechanical performance of muscles strongly affects performance. Human competitive athletic sports revolve around this idea: behavioural performance reflects the physical muscle ability of an athlete. Since it is muscle exercise that changes muscle properties, behavioural performance in sports reflects recent exercise status. Our paper extends this idea to vocal behaviours.

In our singing prevention paradigm, we manipulate the activity of singing behaviour by exploiting the fact that zebra finches do not sing in the dark. Our manipulation causes reduction of singing activity and thus reduction of muscle exercise. We observe changes in muscle performance that are consistent with reduced activity and in the same direction as completely abolishing muscle activity by denervation. As detailed above, we can exclude stress as a major factor. Birds that are kept in the dark for extended periods of time are expected to experience prolonged melatonin exposure that have been shown to efficiently *counteract* muscle atrophy in humans and mammalian animal models (Oner et al. 2008; Stacchiotti, Favero, and Rodella 2020). During this dark period singing motivation may have been high, but the birds did not execute singing behaviour in the dark. Thus, stress, motivation and melatonin cannot explain the changed muscle phenotype. The observed phenotype is consistent with reduced singing activity.

Based on our current knowledge on vocal production, these changes can be predicted to alter vocal output (as detailed under **Q3**). Based on this we can only confirm our previous conclusion that the acoustic changes we observe were at least in parts, if not fully, caused by reduced muscle performance of the muscles involved in singing.

Aware of other factors that may influence vocal output (such as the ones mentioned by the reviewer), we excluded the cues motif duration and source level in our female choice test. The majority of females still very clearly discriminated between the two versions of a motif they were

presented with and 80% chose the song produced before the singing prevention paradigm. We think that the female choice experiment shows that vocal output contains information about the producer's state. Additionally, we have shown that syringeal muscles atrophied due to singing prevention and that the vocal changes we observed are consistent with loss of muscle performance. Together we thus still think that our results support our initial conclusion.

I. 36: Here the premise of the paper is stated. Neither part of the manuscript really tests this question directly and unambiguously

We respectfully disagree. In our manuscript we use a reductionist framework: we aim to study the effect of a feature on a property by removing that feature, in this case muscle exercise. By removing muscle exercise, we can study the effects of exercise.

Hypothesis 1: Vocal muscles require exercise to achieve and maintain adult performance.

We test and provide clear evidence that removal of vocal muscle exercise prevents achieving and maintaining adult performance. We use denervation to prevent muscle exercise and show that the intact, but denervated muscle does not develop or maintain the performance characteristics (FWHM, MIS and CSA) we observe in exercised adult muscle. Denervation is the most extreme intervention to prevent exercise. It would be interesting to see if adult birds can be made to exercise more, but since the muscles are already performing at the maximal speed of any synchronous muscles, this seems unlikely, and we suggested an alternative experiment (**Line 621-625**). As a milder and more specific muscle use prevention, we performed the singing prevention experiment that allowed the muscles to be active during breathing and calling. We show that muscle performance changes in the same direction as short and long-term denervations. Thus, without exercise muscle performance alters, and logically the exercise is thus causing the performance change.

Hypothesis 2: Changes in vocal muscle performance affect vocal output.

As discussed in detail in **Q3**, we agree and are aware of the fact that vocal output in our in our singing prevention paradigm also changed due to effects unrelated to changes in muscle performance. However, we argue that some of the changes (e.g. the effect on fundamental frequency) are consistent with and can only be explained by the reduced muscle performance we measured in these animals. We included a paragraph in the discussion detailing our argumentation (**Line 498-523**).

I. 55: Descriptions of changes after denervation merely confirm what has been found earlier (Julie Wade's work).

We think that this comment stems from a misunderstanding due to our unclear phrasing and have rewritten the sentence to convey our main message more clearly.

Main text (Line 108-115): "We compared these muscle features on both sides when these animals were adult (100 days post hatching) and song learning was completed. Denervation halved the contraction speed (FWHM doubled from 4.89±0.86ms to 9.13±3.18ms, unpaired two-sided Welch's t-test, p=0.00208, N=7), quadrupled MIS from 2.61±1.37mN/mm² to 12.23±9.36mN/mm² (Unpaired two-sided Welch's t-test, p=0.01508, N=7; Fig 1B), and reduced cross-sectional area (CSA) by 70% (Fig S1, paired two-sided Welch's t-test, p=0.00241, N=4). The myofibre number remained the same (Fig S1, paired two-sided Welch's t-test, p=0.04385, N=4). Treatment did not affect the intact side, because both contraction speed and MIS were not different from wildtype adult males (Fig 1B, Data S1)."

We carefully checked all papers published by Julie Wade and colleagues. We could only find that they performed denervation in one of their publications to identify motor neurons innervating syringeal muscles (Wade and Buhlman 2000). They quantified muscle fibre diameter (not area nor number) on histological cross-sections and syringeal mass in these animals, but could not find a

statistically significant effect of the denervation. This may have been due to the low number of fibres quantified (25 versus ~30,000 in our study) as well as the design of their statistical test.

However, more importantly, we are interested in the mechanical performance of muscle tissue, which can only be established by measuring force using physiological methods and the area of muscle fibre bundles. As far as we know, all published data on mechanical muscle performance of syringeal muscles to date has been acquired and published by our lab. We are confident that our findings are novel and not a replication of existing data.

I. 74: Again, these data do not show that daily exercise is required. They show that denervation leads to the documented changes – but that is very different from how daily exercise may maintain muscle phenotype.

We cannot conclude to have established that exercise (as loading paradigm) has the same strength as denervation (as unloading paradigm). However, that these paradigms have opposite effects which is well-established in the muscle exercise literature (see **Q1** and **L36** question above). By removing the effect strength, we have toned down this concluding sentence into:

Main text (Line 131-132): “Taken together, these data support the hypothesis that maintaining adult peak muscle performance requires vocal muscle exercise.”

I. 189: This conclusion is not merited from the presented data (see major points).

This comment refers to the sentence: “*Thus, short-term singing prevention affects speed and MIS in the same direction as short-term denervation.*”

Because we don’t think the reviewer is questioning the results that speed and MIS are affected, we think it must be related to the naming of singing prevention as the major cause. To make a more clear distinction between the alternative changes (melatonin, stress, motivation; See **Q3**) that our paradigm to prevent song could have induced, we have rephrased this sentence into:

Main text (Line 337-339): “Because the short-term singing prevention paradigm affects speed, MIS and proteome in the same direction as short-term denervation, it acts as a milder unloading paradigm compared to nerve cuts.”

I. 196: Does the decrease in frequency correspond to the decrease in amplitude?

We have investigated our data and do not see any evidence for systematic correlation between f_0 and source level. We agree that this is a very interesting question, but because of space limitation we decided to not include this analysis in our revised manuscript.

I. 200: Again, there are other possible explanations for the observed changes. This conclusion is therefore not unequivocally tested.

We thank the reviewer for their comment. As mentioned under **Q3** we have added discussion on this topic. We have rephrased this sentence into:

Main text (Line 348-350): “Taken together, our acoustic analyses showed that only one week of our targeted singing prevention paradigm causes significant shifts in vocal output.”

I. 203: Yes, but more recent work directly on zebra finches (and not Bengalese finches) has added to our knowledge.

We apologize for our oversight and included the reference to Mendez & Goller (2020) (**Line 507** and **508**).

I. 206: If muscle speed and motif duration decrease, how is this an indication that muscle speed is responsible for the observed changes in song? This conclusion is not valid.

We apologize for unclear phrasing. In our revised manuscript, we have moved this section to the discussion and significantly rephrased this section to better explain our interpretation of the data (**Line 498-523**).

I. 211: The temperature connection is only one possible interpretation. Melatonin may play a part, as does motivation. Again, this is a very selective use of different aspects known to affect motif duration.

Again, we apologize that our discussion of the effects of our singing prevention paradigm was incomplete. We hope we have answered this question at your query **Q2** and **Q3** above. We have moved this paragraph to the discussion and expanded it (**Line 498-523**). Furthermore, we added a new paragraph in our discussion to cover alternative explanations including melatonin and motivation (**Line 480-498**).

Discussion: Because of the many concerns with the data sets and their interpretations, most of the points raised in the discussion are less clear than stated. This needs to be toned down in light of the many reservations about the disjunct parts.

In our revised manuscript, we have significantly rewritten both introduction and discussion. We have added paragraph on the alternative explanation as detailed in our answers to **Q1 - Q3** above.

Point-by-point response to reviewers NCOMMS-23-12907

Reviewer #2

This is a very neat discovery and a beautifully executed set of experiments. I have no doubt this will be a much-cited, significant contribution to the literature in this field, as well as broadly relevant (and interesting) outside of the songbird or vocal production world. I do not have major criticisms of the science, but only a few minor comments and suggestions.

Generally, the manuscript is extremely dense, which is understandable given the amount of work presented here, but in the relatively short format for this journal, it makes clarity a bit of an issue.

Abstract

Aware of the very short format of abstracts for the journal, but the result is a few sentences which are not clear:

Q1: line 17 "...it is unknown whether vocal, like limb muscles, exhibit..." ideally this should read "vocal muscles, like limb muscles" as the reader may not be aware that there even are vocal muscles. I had to read it a couple of times to clock that "vocal" modified "muscles", and rather thought that the authors had just left off a word somewhere.

A1: We thank the reviewer for pointing this out. We have rephrased this as suggested.

Abstract (Line 15-17): "While the acquisition and maintenance of motor skills generally requires practice to develop and maintain both motor circuitry and muscle performance, it is unknown whether vocal muscles, like limb muscles, exhibit exercise-induced plasticity."

Q2: Line 19 Similar issue with "Preventing singing rapidly alters" where it is not clear if the authors mean that the prevention of rapid singing alters muscles, or if the prevention of singing results in rapid alteration of muscles.

A2: To avoid confusion we have removed "rapidly" and rephrased into:

Abstract (Line 18-19): "Preventing singing alters both muscle and vocal performance within days and females prefer song of vocally exercised males."

Results

Q3: Line 47, not sure about use of "vocal folds" here as the vibratory structures in the oscine syrinx are not really vocal folds (despite similar function).

A3: We fully agree and apologize for this oversight. We have rephrased into "vibrating labia" (Line 101).

Q4: Line 53, would be very helpful to give a functional definition of maximal isometric stress here. I think for the lay-biologist it is clear what twitch contraction and contraction speed would mean both in terms of muscle behaviour, but MIS is more oblique and its relevance may be lost for much of the intended readership (especially considering how central this change is to your main findings).

A4: We have added explanatory graphs into **Fig 1B** clearly illustrating the FWHM and MIS measurements. Furthermore, we have added clear definitions and rephrased this section into:

Main text (Line 104-108): "Next, we measured two key features of physiological muscle performance in the dorsal tracheosyringeal muscle: 1, contraction speed, measured as the full width at half maximum (FWHM) of force development during twitch contraction, and 2,

maximal contraction force per cross-sectional area, i.e. the maximal isometric stress (MIS) during tetanic contraction (Fig 1B).”

Q5: This is also so for the legend of Fig 1, where the functional definition of FWHM is not provided until deep in the methods at the end of the paper. I think readers that are not muscle physiologists will struggle to put together the significance of these measures as the manuscript is currently presented.

A5: We thank the reviewer for pointing this out. We have added the definition of FWHM in the main text (see previous **Q4**). Furthermore, we added explanatory graphs into **Fig 1B** clearly illustrating the FWHM and MIS measurements.

Q6: Line 195, were these changes statistically significant between pre- and post- treatment birds? For example, source level difference does not seem particularly convincing given the size of the errors? Edit- I see some p-values in Fig 3 to support this.

A6: We apologize for unclear phrasing. To increase readability and reduce the number of words in the main text, we initially chose to place all statistical test details of the main text and figures in the included excel file **Data S1**. However, in our revision, we placed basic information of test results back into the main text and figures throughout.

Also, we weren't sure if the reviewer was also referring to the sentence “*Singing prevention caused several acoustic changes to song*”. This sentence was meant as introduction to the following sentences describing those differences. To clarify, we have rephrased this section into:

Main text (Line 342-346): “Singing prevention caused several acoustic changes to song. First, motif duration decreased from 0.69 ± 0.15 to 0.67 ± 0.14 s (Paired two-sided t-test, $p = 7.7 \times 10^{-5}$, $N = 10$), i.e. decreasing by $3.7 \pm 1.4\%$ and motif source level decreased from 51.7 ± 2.0 to 50.1 ± 2.9 dB SPL at 1m (Fig 3C-F) (Paired two-sided t-test, $p = 0.011$, $N = 10$). Second, we extracted time-resolved fundamental frequency (f_0) traces over individual syllables within motifs (Fig 3G).”

Q7: Regardless of statistical significance, the behaviour experiment here would suggest that these changes are biologically relevant, I suppose. It is amazing that females seem to show such a strong preference for what seem to me to be relatively minor changes. Do the authors have ideas about which of these changes seem more salient to females?

A7: We agree with the reviewer and were very excited to see this. In our experiments we excluded the two cues that have been established to affect female choice, namely motif duration and source level, and still get a strong preference. These data lend further support to recent work by Bob Dooling et al who showed that finches can discriminate between very small temporal and spectral differences in acoustics. Our work now further established they also base critical mating decisions on these changes.

At this point we do not know for sure what cues the females use and this is the focus of our current research plan. Fundamental frequency is one of many candidates, but we need more repetitions of the female choice experiment to be conclusive.

Q8: What is the point of the “duration decrease” bar in Figure 2, panel F? Isn't this information already in the left graph of panel 3F? is a graph really necessary there? Perhaps this needs a clearer explanation.

A8: We agree with the reviewer. We have omitted the bar in **Fig 3F** and kept the results in the main text.

Q9: Figure 3, panel D, again this just needs a bit more description. What goes into the colour mapping of each row in the aligned motifs? Is this a measure of consistency across multiple repeats of the motif in each condition?

A9: Yes, each row is a single motif iteration with 284 motifs/rows before and 201 after singing prevention. The plot shows how beautifully and strikingly stereotyped the singing behaviour is for readers not familiar with the songbird system, while displaying all raw data of a single individual.

To clarify, we indicated the alignment location, removed abbreviations in the figure and rephrased the figure legend into:

Fig 3 legend (Line 384-386): “D, Motif iterations pre (n=284) and post (n=201) singing prevention of same individual aligned to the onset of syllable three (downward arrow). Each row represents a single motif iteration color-coded for source level (top), f_0 (middle) and Wiener entropy (bottom).”

Discussion

Q10: I am curious as to why calling would not be adequate to exercise syringeal muscles, and what this means for female syringeal development and maintenance?

A10: This is a very intriguing aspect of our findings that we are looking into. One likely possibility we propose is that the motoneurons are firing at higher rates during song compared to calls in both sexes and that the lower rates during calls is below a threshold to increase MYH13 expression. Unfortunately, there is no recordings of nXIIIts neurons during these behaviours to determine firing rates. The closest data we have is firing rates of neurons in premotor nucleus RA and these are higher during song (Yu and Margoliash, 1996), which is consistent with this idea (**Line 471-474**).

Although definitely answering this question is outside the scope of this manuscript, we have added a paragraph in the discussion that specifically addresses implications and predictions for female syringeal development (**Line 536-547**).

Q11: Do the syrinxes of females (in species without female song) remain in a paedomorphic condition? Or is there a maturation, and would a female prevented from calling be expected to lose muscle mass and contractile speed?

A11: We think our paper opens for lots of these interesting questions and we hope the added paragraph in the discussion (see **Q10** above) covers some of these.

Methods

Q12: Line 347, please include a note of how far cranially/caudally the section of NXIIIts was removed (or did this vary between individuals?)

A12: We have added this information.

Methods (Line 734-736): “We removed at least 5mm of the right branch by cutting the nerve with a pair of micro-scissors from about 10 to 15 mm cranial from the syrinx.”

Q13: Line 302-4, this is an interesting idea, but it is not clear how the results presented here actually support it (other than contraction speed decreases with disuse), is the only difference

between superfast and fast twitch fibres simple a matter of usage? Could the authors provide a bit more clarity in what the hypothesized mechanism would be here?

A13: We apologize that this section ended up not as clear as intended. Indeed, our data shows that contraction speed decreases with disuse. We furthermore show that CSA (**Fig 2CD**) and mitochondrial function (**Fig 2F**) decrease with disuse.

The current general consensus on the neural regulation of muscle fibre type in limb muscles is that manipulations can be roughly divided into loading and unloading paradigms. Thus indeed, muscle fibres shift type dependent on use.

Based on our data, we observed two things. Firstly, that the direction of neural regulation of speed in syringeal muscles is opposite from limb muscles. Secondly, that laryngeal muscles also have this opposite speed regulation. Thus, we raise the hypothesis that laryngeal and syringeal muscles react in the same direction, but in an opposite way to limb muscles. This is important because the state-of-the-art human voice training paradigms that focus on altering muscle properties have based their training predominantly on limb muscle physiology, which is thus leading to opposite effects.

Next, we speculate that the similarity between laryngeal and syringeal muscles is probably due to myogenic factors: they both belong to the craniofacial muscle group. The molecular mechanisms underlying fibre type regulation in limb muscles have been largely uncovered, but remain understudied in the craniofacial muscles, including laryngeal and syringeal muscles, and needs further investigation.

In our revision, we now added the topic in the introduction (**Line 44-61**) and rephrased this section in the discussion for clarification (**Line 501-524**).

Q14: Line 312, is there a missing (or redundant) word here: “A complete lack of both access and experimental access to human laryngeal...”

A14: We have rephrased this section into:

Main text (Line 525-526): “A lack of experimental access to human laryngeal muscular function in vivo, has heeded the explicit call for animal models that can be trained in paradigms similar to human vocal training.

Q15: Line 321, I agree that these results are really broadly relevant and can aid in better understanding motor plasticity in response to training. Nonetheless the differences in the tissue composition of mammalian vocal folds and syringeal vibratory masses I think make this final claim a bit too bold, as mammalian vocal disorders are often linked to pathologies of the vocal folds themselves (which are themselves muscular and vascularised in a way that avian syringeal labia are not), rather than the laryngeal muscles that control their position and tension.

A15: We apologize for the unclear phrasing and agree with the reviewer. We intended to write that we think that songbirds are a great model for studying pathologies regarding *vocal muscles*, and not generally voice, and indeed not vocal *fold* pathologies. We have rephrased this section into:

Discussion (Line 534-536): “Therefore, we propose songbird as a powerful animal model to study pathophysiology of human voice disorders concerning laryngeal muscle function and for further development of therapeutic interventions and pedagogical tools for voice therapy and (re)habilitation of laryngeal muscles.”

Supplementary Text

Q16: The authors state here that it is still not know how the two muscle fibre types differ in their responses to exercise. In the histological investigation of muscle cross sections, it is not possible to compare responses to denervation by superfast and normal fast twitch fibres, and compare for example relative decrease in diameter at the minimum?

A16: We apologize for unclear phrasing. In our experimental design we cannot follow the response of individual fibres to the nerve cut, because the histology requires us to sacrifice the muscle. We do quantify fibre area of all individual fibres (**Fig 2C**), but can only make statements on how the fibre type population is distributed before and after. We propose a model how individual muscle fibres respond to unloading (**Fig S3**), but this model needs to be tested. That said, we are not aware of any muscle physiology study able to identify and follow individual muscle fibres over interventions just as neuroscientist are able to follow the same neurons in the brain. That is a very cool idea to pursue.

We have rephrased this into:

Supplementary text: “Songbird syringeal muscles contain two fibre types; fast and superfast (Mead et al. 2017; Christensen et al. 2017), but is currently not known if, and how individual fibres of each fibre type respond to exercise paradigms.”

Q17: General comment, please be consistent in either British spelling or American spelling (e.g. fiber and fibre both occur in current text)

A17: Thank you, we have changed to British spelling according to journal requirement.

Point-by-point response to reviewers NCOMMS-23-12907

Reviewer #3

Summary

I have reviewed the paper “peak performance singing requires daily vocal exercise in songbirds and found it to be technical sound to the best of my knowledge,

Q1: the presentation of the research question and the objectives and hypotheses are not clearly stated, nor is there a lay summary of the approach leading into the technical results section which would be helpful.

The extremely brief introduction is inadequate in its goal of introducing the reader to the overall question, hypotheses, objective, and approach and therefore the rest of the manuscript is difficult to navigate and further the importance and implications of this carefully done work are not apparent. I recommend the authors make a more thorough attempt at a concise yet useful introduction.

Introduction: Even for a short format journal like Nature communications this introduction is inadequate. It fails to communicate the context of this study, the basis for the hypotheses, or the timelines of the approach.

A1: In our revised manuscript we have completely rephrased our introduction to improve it on the abovementioned points. We did not include it in the rebuttal, but refer to the track changes document (**Line 33-74**).

Results:

Q2: The results are clearly presented, although the use of acronyms for a general audience journal is excessive. Consider limiting the number of acronyms when possible.

A2: We thank the reviewer for and have reconsidered our use of acronyms. We kept the following three uncommon abbreviations that we consider essential to the manuscript: MIS, FWHM and f. The abbreviation CSA for cross-sectional area is so commonly used that we also kept this.

In our revised manuscript we removed SFM, DPH, WE, SL, DTB, and RA in the main text and figures. The abbreviation DEN we did not use in the text, only figures and was removed where possible.

The gene/protein (eg, MyHCs, MYH13) and antibody names (e.g. MY-32), are not acronyms but proper names and were therefore kept.

Q3: The statistics section indicates that the details of the statistical tests are presented for each section of the results - this is not the case. Please clearly indicate the statistical approach used for each set of experiments somewhere accessible in the main manuscript.

A3: We apologize for this unclarity. We initially chose to place all statistical test details of the main text in figures in the included excel file **Extended Data S1** to increase readability and reduce the number of words in the main text. However, in our revision, we added the most basic information of test results back into the main text and figures.

Q4: Discussion: Overall the discussion is thoughtful but limited in scope. Can the authors expand the scope to include more ecological/evolutionary or bigger picture implications of these findings?

A4: In our revision we have extended the discussion significantly (**Line 480-523**) and added a new paragraph speculating on effects on female development on ecological and evolutionary context (**Line 536-547**).

Q5: Also more discussion on all the changes in proteins etc highlighted in fig 2 would be helpful.

A5: Due to other reviewer requests, we have added significant text sections to introduction and discussion. Therefore, we decided not to include more discussion on this topic. Luckily, we have manuscripts close to submission that are focussed only on proteomics of these muscles, including discussions on function.

In our revised manuscript we rephrase the concluding sentence of this section into:

Main text (Line 254-257): “Thus, denervation reduced overall abundance of protein categories that set muscle speed, such as calcium handling and mitochondrial function, and drove composition change of MyHCs from fast to slow isoforms, consistent with our morphological and physiological data.”

Q6: The data in fig 3 seems particularly interesting, but it is all but overlooked.

A6: In our revised manuscript, we moved the paragraph discussing the effects of our singing prevention paradigm to the discussion. We significantly rephrased the paragraph to discuss the data in **Fig 3 (Line 498-523)** and added a new paragraph discussing possible alternative explanations of our singing prevention paradigm as requested by Reviewer#2 (**Line 480-497**).

Q7: Overall the authors have conducted a technically sound study, but the manuscript does not do it justice, especially for a broad scientific audience.

A7: We sincerely hope that the numerous changes we made in our revision have significantly improved the manuscript, so that it speaks even better to a broad audience.

- Christensen, Linsey A., Lisa M. Allred, Franz Goller, and Ron A. Meyers. 2017. 'Is sexual dimorphism in singing behavior related to syringeal muscle composition?', *The Auk*, 134: 710-20.
- Daisuke, Mizuguchi, Sánchez-Valpuesta Miguel, Kim Yunbok, B. dos Santos Ednei, Kang HiJee, Mori Chihiro, Wada Kazuhiro, and Kojima Satoshi. 2023. 'The song does not remain the same: daily singing of adult songbirds prevents passive changes in song structure independently of auditory feedback', *bioRxiv*: 2023.02.22.529516.
- Mead, A. F., N. Osinalde, N. Ortenblad, J. Nielsen, J. Brewer, M. Vellema, I. Adam, C. Scharff, Y. Song, U. Frandsen, B. Blagoev, I. Kratchmarova, and C. P. Elemans. 2017. 'Fundamental constraints in synchronous muscle limit superfast motor control in vertebrates', *Elife*, 6.
- Ohlendieck, K. 2011. 'Proteomic profiling of skeletal muscle plasticity', *Muscles Ligaments Tendons J*, 1: 119-26.
- Oner, J., H. Oner, Z. Sahin, R. Demir, and I. Ustunel. 2008. 'Melatonin is as effective as testosterone in the prevention of soleus muscle atrophy induced by castration in rats', *Anat Rec (Hoboken)*, 291: 448-55.
- Stacchiotti, A., G. Favero, and L. F. Rodella. 2020. 'Impact of Melatonin on Skeletal Muscle and Exercise', *Cells*, 9.
- Wade, J., and L. Buhlman. 2000. 'Lateralization and effects of adult androgen in a sexually dimorphic neuromuscular system controlling song in zebra finches', *J Comp Neurol*, 426: 154-64.

REVIEWERS' COMMENTS

Reviewer #1 (Remarks to the Author):

In this revised version the authors have toned down the points of contention of the original version. There are a few issues that remain in my opinion.

- 1) The reference to the mammalian muscle disuse literature is highly selective and does not reflect the wealth and breadth of information (e.g., l. 36-39; discussion).
- 2) It would be interesting to compare results to recent studies in mammals (e.g., single fiber changes in a mammalian system; Lang et al., J. Proteome Res. 2018, 17, 3333–3347). It is likely that there are also other relevant studies.
- 3) The metabolic cost of birdsong production is incorrectly stated. Zollinger et al. found that they could not measure an increase in oxygen consumption over that of 'normal' amplitude when zebra finches sang more loudly (induced by Lombard effect). However, there is a measurable cost to singing in birds (including zebra finches) - albeit only a small one (e.g. for zebra finches: Oberweger et al., 2001; Franz and Goller, 2003).

A small comment:

l. 83: I would not call untreated animals 'wildtype'

Reviewer #2 (Remarks to the Author):

This revised manuscript has done an excellent job at addressing the major comments and suggestions from the first round of reviews.

As before, i think this is a really tight, neat and exciting experimental study.

I think the additional text, changes to wording to make claims a bit more conservative and the addition of the proteomics work really contributed to improve this work.

While the method the authors used to prevent vocalisations (or song) by keeping birds in the dark would indeed be likely to result in other behavioural or physiological effects as well (e.g. myriad hormonal shifts at the very least), I do not believe that these would preclude being able to make conclusions on the impact of disuse on syringeal muscle. Thus, i find the additional paragraph provides sufficient evidence on predicted outcome of hormonal shifts to reassure one about the conclusions of the authors.

Likewise, the addition of the proteomics work (Supplementary fig 2) I believe does show that disuse (either by denervation or singing prevention) leads to the loss in vocal motor performance.

a couple of very minor comments:

line 45 (in the merged new ms file), it's not clear what is meant by "access" in "but we lack in vivo access". Can this be reworded or clarified? Is the access to do with available imaging tools? That the muscles in question can't be observed endoscopically? Other?

Line 85, 94, you compare denervated syrinx to "wildtype adult male", here I assume you mean "intact" or "unmanipulated" adult male? Or do you really mean wildtype in the genetic sense (not domesticated)? And, related to that, would make sense to specify which zebra finch species used, by referring to latin name, in the first mention of the species in the ms, unless this is somehow not the style of this journal? The species name provided in the methods is the Sunda zebra finch (*T. guttata*) rather than Australian (*T. castanotis*), is that correct? Were all the birds used from a domesticated lab-bred strain?

Reviewer #3 (Remarks to the Author):

I find the manuscript much improved and all my initial critiques have been addressed.

Point-by-point response to reviewers NCOMMS-23-12907

We thank the reviewers for their effort to thoroughly and carefully review our paper, and for the positive and constructive comments on our manuscript. In the revised manuscript we have addressed the concerns that they have raised, as described below. We have included short sections of rephrased text in the rebuttal with new text underlined and refer to line numbers in the revised *track changes* document. To improve readability and avoid repetition, we did not include longer new sections in the rebuttal, but refer to line numbers only.

Reviewer #1

REVIEWERS' COMMENTS

Reviewer #1 (Remarks to the Author):

In this revised version the authors have toned down the points of contention of the original version. There are a few issues that remain in my opinion.

Q1: 1) The reference to the mammalian muscle disuse literature is highly selective and does not reflect the wealth and breadth of information (e.g., l. 36-39; discussion).

A1: While we agree that there is a large body of literature, we respectfully disagree that we cite selective. We cite two reviews that are highly regarded in the field, one specific to muscle physiology and one specific to muscle exercise, with together more than 2000 references to original work. We think these present a balanced representation of the knowledge in the field.

Q2: 2) It would be interesting to compare results to recent studies in mammals (e.g., single fiber changes in a mammalian system; Lang et al., J. Proteome Res. 2018, 17, 3333–3347). It is likely that there are also other relevant studies.

A2: We thank the reviewer for this suggestion and have now added this reference to the discussion regarding the opposite effect on muscle speed we observe in vocal muscle versus limb muscle. We have searched but are not aware of other studies looking at proteomic data of disused vocal muscles.

Line 420-423: "In well-studied limb muscles, loading paradigms increase CSA and mitochondrial function, and generally transform fast into slower fibre types(Lang F et al., 2018;Ohlendieck K, 2011), while unloading paradigms have the opposite effect, decreasing CSA and mitochondrial function and transforming slow into faster, more fatigable, fibre types (**Supplementary Fig 3**)."

Q3: 3) The metabolic cost of birdsong production is incorrectly stated. Zollinger et al. found that they could not measure an increase in oxygen consumption over that of 'normal' amplitude when zebra finches sang more loudly (induced by Lombard effect). However, there is a measurable cost to singing in birds (including zebra finches) - albeit only a small one (e.g.for zebra finches: Oberweger et al., 2001; Franz and Goller, 2003).

A3: We have rephrased the paragraph to better reflect the state of the art:
Line 391-394:b"Our data strongly suggest that song exercise is a previously unrecognized cost of adult song maintenance. Such costs are expected for sexually selected signals, but in birdsong direct metabolic costs of singing are low(Franz M and Goller F, 2003;Oberweger K and Goller F, 2001;Zollinger SA et al., 2011). Consequently, with the low physiological costs to produce song, costs were considered mostly

developmental and song an honest indicator of past condition (the developmental stress hypothesis(Nowicki S and Searcy WA, 2004)).”

A small comment:

Q4: l. 83: I would not call untreated animals 'wildtype'

A4: We changed our wording to „untreated“.

Line 93-94: “Treatment did not affect the intact side, because both contraction speed and MIS were not different from untreated adult males (**Fig 1B, Supplementary Data 1**).”

Line 103-105: “The effects of denervation on MIS were even more dramatic: merely two days post denervation MIS dropped fivefold compared to the intact side (from 7.13 ± 4.85 to 1.43 ± 1.88 mN/mm², unpaired Welch’s t-test, $p=0.00148$, **Fig 1F**), which remained unaffected compared to untreated males.”

Reviewer #2 (Remarks to the Author):

This revised manuscript has done an excellent job at addressing the major comments and suggestions from the first round of reviews.

As before, i think this is a really tight, neat and exciting experimental study.

I think the additional text, changes to wording to make claims a bit more conservative and the addition of the proteomics work really contributed to improve this work.

While the method the authors used to prevent vocalisations (or song) by keeping birds in the dark would indeed be likely to result in other behavioural or physiological effects as well (e.g. myriad hormonal shifts at the very least), I do not believe that these would preclude being able to make conclusions on the impact of disuse on syringeal muscle. Thus, i find the additional paragraph provides sufficient evidence on predicted outcome of hormonal shifts to reassure one about the conclusions of the authors.

Likewise, the addition of the proteomics work (Supplementary fig 2) I believe does show that disuse (either by denervation or singing prevention) leads to the loss in vocal motor performance.

a couple of very minor comments:

Q1: line 45 (in the merged new ms file), it's not clear what is meant by "access" in "but we lack in vivo access". Can this be reworded or clarified? Is the access to do with available imaging tools? That the muscles in question can't be observed endoscopically? Other?

A1: We have rephrased to

Line 48-50: “In humans, the larynx and its muscles are hypothesized to change with training or age, but in vivo experiments are challenging or impossible (Johnson AM and Sandage MJ, 2021).”

Line 443-444: “Experimental challenges in studying human laryngeal muscular function in vivo, has heeded the explicit call for animal models that can be trained in paradigms similar to human vocal training(Johnson AM and Sandage MJ, 2021).”

Q2: Line 85, 94, you compare denervated syrinx to "wildtype adult male", here I assume you mean "intact" or "unmanipulated" adult male? Or do you really mean wildtype in the genetic sense (not domesticated)?

A2: We changed our wording to „untreated“.

Line 93-94: “Treatment did not affect the intact side, because both contraction speed and MIS were not different from untreated adult males (**Fig 1B, Supplementary Data 1**).”

Line 103-105: “The effects of denervation on MIS were even more dramatic: merely two days post denervation MIS dropped fivefold compared to the intact side (from 7.13 ± 4.85

to $1.43 \pm 1.88 \text{ mN/mm}^2$, unpaired Welch's t-test, $p=0.00148$, **Fig 1F**), which remained unaffected compared to untreated males."

Q3: And, related to that, would make sense to specify which zebra finch species used, by referring to latin name, in the first mention of the species in the ms, unless this is somehow not the style of this journal?

A3: We have now added the latin name at first mention in the manuscript.

Line 17-19: "Here, we show that juvenile and adult zebra finches (*Taeniopygia castanotis*) require daily vocal exercise to first gain and subsequently maintain peak vocal muscle performance."

Q4: The species name provided in the methods is the Sunda zebra finch (*T. guttata*) rather than Australian (*T. castanotis*), is that correct? Were all the birds used from a domesticated lab-bred strain?

A4: We onyl used domesticated lab-bred animals and have corrected the species name to (*Taeniopygia castanotis*).

Reviewer #3 (Remarks to the Author):

I find the manuscript much improved and all my initial critiques have been addressed.